# Advanced parametrization for the production of high-energy solid-state lithium pouch cells containing polymer electrolytes

Wonmi Lee [1,7], Juho Lee [1,2,7], Taegyun Yu [1,7], Hyeong-Jong Kim [2], Min Kyung Kim [1,3], Sungbin Jang [1], Juhee Kim [1], Yu-Jin Han [1], Sunghun Choi [4,5], Sinho Choi[1], Tae-Hee Kim [1], Sang-Hoon Park [1], Wooyoung Jin [1], Gyujin Song [1], Dong-Hwa Seo [6], Sung-Kyun Jung [2] ✉ & Jinsoo Kim [1] ✉

Lithium batteries with solid-state electrolytes are an appealing alternative to state-of-the-art non-aqueous lithium-ion batteries with liquid electrolytes because of safety and energy aspects. However, engineering development at the cell level for lithium batteries with solid-state electrolytes is limited. Here, to advance this aspect and produce high-energy lithium cells, we introduce a cell design based on advanced parametrization of microstructural and architectural parameters of electrode and electrolyte components. To validate the cell design proposed, we assemble and test (applying a stack pressure of 3.74 MPa at 45 °C) 10-layer and 4-layer solid-state lithium pouch cells with a solid polymer electrolyte, resulting in an initial specific energy of 280 Wh kg$^{-1}$ (corresponding to an energy density of 600 Wh L$^{-1}$) and 310 Wh kg$^{-1}$ (corresponding to an energy density of 650 Wh L$^{-1}$) respectively.

Li-ion batteries (LIBs) have been widely used for applications ranging from portable mobile devices to electric vehicles owing to their high specific energies[1,2]. However, the risk of thermal runaway stemming from the flammable liquid electrolyte and cell degradation remains a concern[3]. Therefore, solid-state batteries (SSBs), which replace the liquid electrolyte with a solid electrolyte (SE), are considered promising alternatives[4,5]. SSBs offer several potential advantages over conventional LIBs: 1. SEs are typically non-flammable, ensuring safety even when unexpected cell shorting occurs[6]. 2. Li metal can be used as a negative electrode material if the ratio between the Li partial molar volume ($V_{SE}/V_{Li}$) versus shear modulus ($G_{SE}/G_{Li}$) of the SE is optimized[7–9]. 3. It can potentially enhance the system-level specific energy (energy per mass) through efficient packaging design such as the use of a bipolar stacking structure[10]. Despite the potential for high

specific energy, a systematic design approach for high-loading-density electrodes and multi-stacked cell manufacturing necessary to achieve ultimate high specific energy remains lacking.

Numerous studies have addressed SSB performance improvement in terms of materials-level and lab-scale environments[11–29]; however, the electrode composition is mostly limited to excess amounts of SE and low active material (AM) contents. Most studies use a low AM weight ratio of 30–70 wt%, lower than the required AM content of >90 wt% in conventional LIBs. In addition, a low areal capacity below 3 mAh cm$^{-2}$ and a thick SE layer of over 100 μm serving as the separator are typically encountered in SSB demonstrations, which is far from the ideal cell configuration necessary to attain high specific energy. Hence, it is necessary to conduct systematic research on the design of high-loading electrodes and to establish principles for achieving

[1]Ulsan Advanced Energy Technology R&D Center, Korea Institute of Energy Research, Ulsan, Republic of Korea. [2]Department of Energy Engineering, School of Energy and Chemical Engineering, Ulsan National Institute of Science and Technology (UNIST), Ulsan, Republic of Korea. [3]Department of Nano Fusion Technology, Pusan National University, Busan, Republic of Korea. [4]Gwangju Clean Energy Research Center, Korea Institute of Energy Research, Gwangju, Republic of Korea. [5]Department of Battery Convergence Engineering, Kangwon National University, Chuncheon, Republic of Korea. [6]Department of Materials Science and Engineering, Korea Advanced Institute of Science and Technology (KAIST), Daejeon, Republic of Korea. [7]These authors contributed equally: Wonmi Lee, Juho Lee, Taegyun Yu. ✉e-mail: skjung@unist.ac.kr; jkim@kier.re.kr

high specific energy SSBs. Several studies have considered various parameters that affect the electrode performance of SSBs, and a few guidelines for electrode design have been proposed[30–35]. These guidelines focus on factors such as the ionic/electronic percolation/conductivity, tortuosity, and AM/SE interfacial area[30,31]. Balancing the ionic and electronic conductivity or controlling the relative particle size of the AM/SE[32,33] have been attempted to improve the percolation. Additionally, maximizing the interfacial ratio of the AM/SE or minimizing the isolated AM particle inside the composite positive electrode have been suggested to promote the utilization ratio of AM particle participating reactions in the electrode[34]. However, uneven contact at the solid–solid interface inevitably increases the sensitivity of each parameter to slight changes in the electrode configuration and loading density[35], thereby creating a trade-off relationship among various factors, making it challenging to suggest a standard electrode design for SSBs.

In addition to electrode design, consideration of cell-level design in achieving high specific energy is scarce. Most research on SSBs has been conducted in a clamped pressurized homemade cell[36] as a proof-of-concept, which is far from the multi-stacked cell design that is a significant advantage of SSBs[37–41]. For SSBs to become commercially viable, it is essential to not only employ a multi-stacked cell design but also to demonstrate cell form factors, such as the prismatic, cylindrical, or pouch types commonly used in conventional LIBs. Although several pouch-type SSBs have been demonstrated, their current level is still limited to a bi-layer configuration (0.6 Ah) with a high specific capacity (> 210 mAh g$^{-1}$) and high areal capacity (>6.8 mAh cm$^{-2}$) using an argyrodite-type SE[6]. Recently, several studies have explored research directions from the materials level to the cell level to improve the characteristics of SSBs for practical proof-of-design[42–44]. However, systematic studies on numerical principles for the design of practical SSBs to achieve higher specific energy at the cell level are elusive.

In this study, we provide rational SSB cell design principles and experimental demonstrations to achieve high specific energy approaching that of commercially relevant LIBs. These guidelines cover possible ideal electrode microstructures as well as cell macro architectures and the accompanying fabrication processes. We built up a multiscale and multi-parameter design space covering electrode composition/density/loading and negative electrode combinations, physical characteristics of the SE membrane, and electrode size/stacking number, all affecting the cell specific energy. Based on this intuitive map and a highly processable solid polymer electrolyte (SPE) as a model SE, a sequential approach was used to construct a 94 wt% AM-containing composite electrode with an areal capacity of over 4 mAh cm$^{-2}$ and an electrode density of ~3.6 g cm$^{-3}$ using isostatic densification. Combining this electrode with a freestanding thin Li metal negative electrode (40 µm) and SE membranes, 10-layered 1 Ah and 4-layered 0.5 Ah scale SSB pouch cells with a specific energy of over 280 and 310 Wh kg$^{-1}$ were successfully built including the whole cell packages.

## Results and discussion
### Design of ideal particle and composite electrode for effective interfacial contact
It is essential to set the composition ratio between the AM and SE in the design of SSB electrodes. There has been considerable prior SSB research; however, no research work has proposed an ideal AM/SE ratio, which is critical in determining the performance characteristics of the SSB at the particle level. To rationally determine the optimal ratio of SSBs, we first made some assumptions: the AM is spherical and has a uniform size without any morphological deformation, and the SE is highly ductile. If we also assume that the AM can be cubic-close packed (CCP), known as the densest interparticle microstructure, then we can derive the remaining interstitial void space as possible occupancy of the SE. To simplify this model for better understanding, we

exclude the amounts of binder and electronically conductive agent as these amounts are negligible relative to the AM and SE contents. Then, we can calculate the ideal composite ratio as 74 vol% AM and 26 vol% SE, as shown in Fig. 1a. Therefore, in this article, we define these values as the "balance threshold", which means that the amounts of AM and SE are well harmonized. Based on this value, the expected SSB properties can be determined toward energy density or power density orientations. The design is energy density oriented in the AM-rich case and power density oriented in the AM-poor case, with most prior SSB research falling into the left side of the balance threshold in Fig. 1a. Even if this threshold is based on an ideal assumption that does not consider the actual particle morphology and size distribution, it is still notable for delivering ideal guidelines for electrode formulation.

From this understanding, the ideal microstructure of the AM/SE composite as an electrode building block was designed and proposed as shown in Fig. 1b, c. To maximize the AM/SE interfacial contact, the necessary use of a core (AM)–shell (SE) structure is apparent if the electron pathway is neglected. Calculating the ionic areal specific resistance (ASR) in the composite particle is also possible based on the known ionic conductivity and physical dimensions such as the shell thickness and radius. The detailed calculation process is presented in Supplementary Fig. 1 and Supplementary Note 1. The ionic ASR through the horizontal pole-to-pole direction tends to increase rapidly because the cross-sectional area converges to zero (point contact) but decreases as the SE is unidirectionally compressed, indicating that the cross-sectional area widens (plane contact). The AM-poor, SE-rich condition represents most recently published research articles (Fig. 1b); however, they must be tailored to have a balanced AM/SE ratio as shown in Fig. 1c to achieve higher specific energy. The trend of the ionic ASR inside AM particles between Fig. 1b, c is inversely proportional to the SE composition because it governs the thickness of the SE shell. To validate this microstructure experimentally, we selected polyvinylidene fluoride-co-hexafluoropropylene (PVDF–HFP)/ lithium bis(fluorosulfonyl)imide (LiTFSI)/ succinonitrile (SN)-based SPE as a model SE due to its high processability. The primary characteristics of the formulated SPE are shown in Supplementary Fig. 2. The SPE exhibits good properties with a high ionic conductivity of $1.08 \times 10^{-3}$ S cm$^{-1}$ at 30 °C and a low activation energy of around 100 meV (Supplementary Fig. 2a, $E_a$) and high lithium transference number of about 0.9 (Supplementary Fig. 2b, $t_{Li+}$). It also exhibits high voltage stability (Supplementary Fig. 2c, >4.9 V vs. Li/Li$^+$), indicating compatibility with high voltage NCM622 (Supplementary Fig. 2d, with a redox potential of NCM622 = 3.76 V) and mechanical robustness (Supplementary Fig. 2e, Young's modulus ≈ 240 MPa) as well as relatively dry characteristics at 25 °C compared to other SPEs (Supplementary Fig. 2f). Moreover, FTIR spectroscopy analysis of the SPE and Li symmetric cycling tests using the SPE were also conducted, with the results presented in Supplementary Figs. 3–5 and Supplementary Notes 2–4.

The key message is the need for a balanced AM/SE ratio and SE ductility to reduce the ionic ASR. However, even if we derive the ideal microstructure of SSBs at the particle level, it is also critical to rationally integrate these unit building blocks at the electrode level. Assuming that these building blocks are also packed in a CCP structure, we can determine the interfacial ionic connectivity using the electrode density as a proxy indicator. We only differentiated the electrode density in maintaining the material composition at the balance threshold. The interfacial microstructures were shown to dramatically change depending on the electrode density (Fig. 1d–g). Figure 1d shows that the interfaces among the core-shell building blocks disconnected from each other at the lower electrode density. A certain level of touching among these core-shell building blocks is necessary in the CCP arrangement, which can be regarded as the so-called "percolation threshold" or the level at which long-range ionic percolation is established by the formation of interfaces among the building blocks (Fig. 1e). Using the same geometric calculation shown

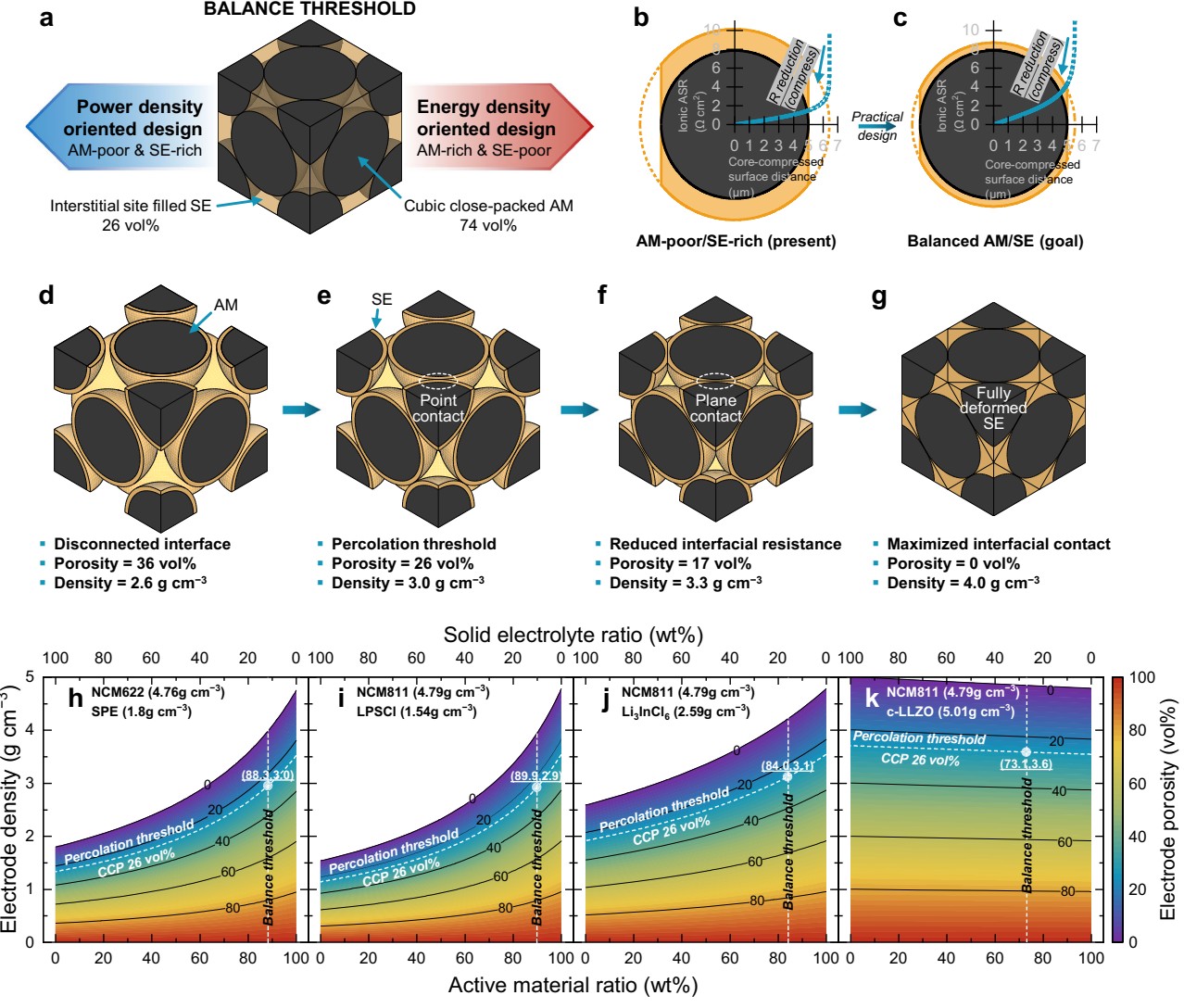

**Fig. 1 | Particle composition and density design strategy for ideal SSB micro-structure. a** Schematic diagram showing the balance threshold between AM and SE for desired characteristics (power density oriented vs. energy density oriented). Ideal core−shell model describing AM/SE composite electrode with maximized interfacial contact and the calculated ionic ASR with 1 mS cm⁻¹ (at 25 °C) SPE under **b** AM-poor/SE-rich condition and **c** balance threshold. **d–g** Schematic diagram representing the different interfacial-contact microstructures related to the electrode density and porosity at identical balance threshold. **h–k** Electrode porosity related to its density as a function of the weight ratio of AM and SE for various types of materials with different true densities. The white dotted lines represent the percolation threshold and balance threshold, respectively, according to the material true densities.

in Fig. 1a, the porosity of the composite electrode should be 26 vol%, and the electrode density would be 3.0 g cm⁻³ if we assume that the AM is NCM622 and the SE is the SPE. At the percolation threshold, the building block interface is a point contact, which possesses infinite resistance owing to its zero-converging cross-sectional area, as shown in Fig. 1b−c; therefore, the composite electrode should be further compressed to have reduced interfacial ionic resistance. In Fig. 1f, the compressed composite with higher electrode density has plane contact at the SE interface, indicating that it would possess reduced interfacial resistance with lower porosity. At the highest electrode density (porosity = 0 vol%) in Fig. 1g, the interfacial contact is maximized, and the resistance would thus be minimized using this structure.

Therefore, designing the desired electrode structure by controlling the AM/SE weight ratio and regulating the electrode density is critical. The true densities of the AM and SE are highly variable; therefore, the balance and percolation thresholds may also slightly change as a function of their combination. Therefore, we plotted several representative combinations of AM/SE composite electrodes in

Fig. 1h−k based on our assumption. Three variables are shown in the contour plot: AM ratio (wt%), electrode density (g cm⁻³), and electrode porosity (vol%). The equivalent balance thresholds (AM = 74 vol%, SE = 26 vol%) converted using their true densities are presented as guidelines. The desired minimum electrode densities are marked and are based on the percolation threshold derived from the electrode porosity as a function of the compositions. From this, we can determine the target electrode compositions and densities according to the SSB design strategies.

Based on this plot with SPE and Li₆PS₅Cl (LPSCl), it is apparent that the SE should be light in terms of the true density to attain higher specific energy. NCM622 and NCM811 as the AM have similar true densities (4.76 g cm⁻³ and 4.79 g cm⁻³, respectively). However, the SE materials considered here have different values (1.80 g cm⁻³ for SPE, 1.54 g cm⁻³ for LPSCl, 2.59 g cm⁻³ for Li₃InCl₆, and 5.01 g cm⁻³ for Li₇La₃Zr₅O₁₂ (LLZO)). The true density is lower than that of the AM for SPE, LPSCl, and Li₃InCl₆. Therefore, the electrode density increases upon increasing the weight fraction of the AM and decreasing the weight fraction of the SE at the same porosity (Fig. 1h−j, the detailed

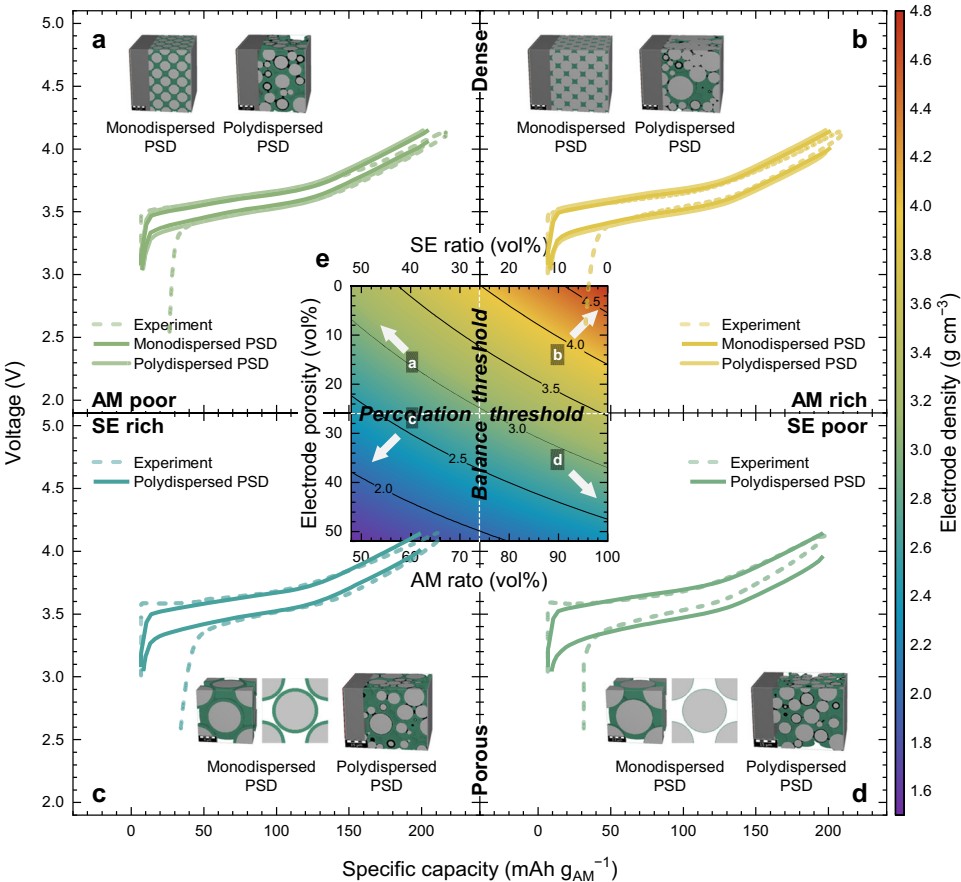

**Fig. 2 | Design principle validation through experimentation and simulations.** Cross-validation of electrochemical characteristics from monodispersed and polydispersed PSD digital-twin models and experimental electrode microstructure. **a** AM-poor and well-percolated, **b** AM-rich and well-percolated, **c** AM-poor and poor-percolated, **d** AM-rich and poor-percolated cases, and **e** converted electrode density landscape of Fig. 1h to represent of balance and percolation thresholds in orthogonal coordinates. Color of voltage profiles indicate the electrode density for each case.

calculation process is presented in Supplementary Note 5). However, LLZO is not considered a better SE because it is usually heavier than NCM-based AMs and because of the difficulty of achieving dense composite electrodes by cold pressing or co-sintering methods[45]. Thus, the electrode density decreases upon increasing the AM weight fraction and decreasing the SE weight fraction at the same porosity (Fig. 1k).

Following the map and baselines of percolation and balance threshold highlighted in Fig. 1h–k, the functionality of the fabricated electrode can be classified depending on the electrode configuration and density. First, below the percolation threshold, electronic and ionic transport would be limited due to the excessive pore-induced poor interface. This is analogous to the SSE fabricated with low fabrication pressure[46]. Above the percolation threshold, the cell performance can be divided into two regimes based on the balance threshold. On the right side (AM-rich, SE-poor regime) and the left side (AM-poor, SE-rich regime) of the balance threshold, the electrode compositions can be determined toward specific energy and specific power orientations, respectively. The poor electronic conductivity or percolation of the AM-poor, SE-rich electrode can be overcome with the addition of an electronically conductive agent, which indicates that the power capability can be improved until the limit of ionic transport. However, the poor ionic conductivity or percolation due to the high tortuosity of the AM-rich, SE-poor electrode is difficult to overcome except through substitution with a more highly conductive SE. Thus, this electrode is more appropriate for high specific energy oriented SSBs. It is thus critical to select a suitable SE

material to achieve the desired electrode porosity and low ionic ASR. This quantitative map can provide guidance for inferring the maximum specific energy at the electrode level as well as the ideal electrode design for a given combination of AM and SE, using an indicator of the AM/SE ratio and electrode density after fabrication.

## High fidelity of design principles verified by simulations and experiment

To validate our electrode design principles, we systematically compared the electrochemical characteristics between experimental and simulated results based on our simplified ideal model and the realistic, complex digital-twin model reflecting particle size distribution (PSD). Validation was representatively proceeded within the condition for each 4 sections of the quadrant in Fig. 1h divided by balance and percolation thresholds as shown in Fig. 2: (a) AM-poor and well-percolated, (b) AM-rich and well-percolated, (c) AM-poor and poor-percolated, and (d) AM-rich and poor-percolated. We used a commercialized BatteryDict program to verify the design principles. As a result, all cases show quite analogous electrochemical voltage profiles and charge capacity between experimental and simulation results as shown in Fig. 2 and Supplementary Fig. 6. We should note slight discrepancies in discharge capacity because BatteryDict was not available to simulate the initial irreversible capacity accompanying such as solid electrolyte interphase (SEI) formation. We point out that the proposed ideal binary model system representing only AM and SE based on the core-shell building block and CCP arrangement does not reflect the electronic percolation network and real particle size

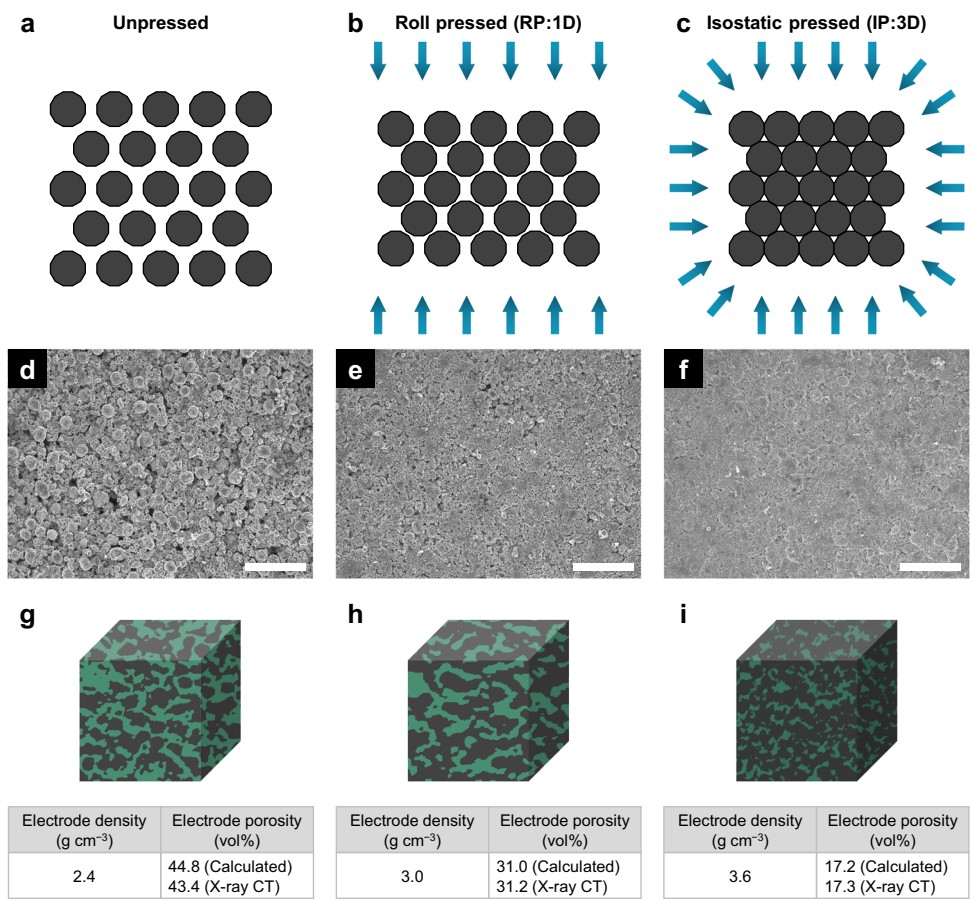

| a | **Unpressed** | b | **Roll pressed (RP:1D)** | c | **Isostatic pressed (IP:3D)** |

| Electrode density (g cm$^{-3}$) | Electrode porosity (vol%) |
| --- | --- |
| 2.4 | 44.8 (Calculated) |
| | 43.4 (X-ray CT) |

| Electrode density (g cm$^{-3}$) | Electrode porosity (vol%) |
| --- | --- |
| 3.0 | 31.0 (Calculated) |
| | 31.2 (X-ray CT) |

| Electrode density (g cm$^{-3}$) | Electrode porosity (vol%) |
| --- | --- |
| 3.6 | 17.2 (Calculated) |
| | 17.3 (X-ray CT) |

**Fig. 3 | Different compression methods and related electrode microstructures.** Schematic diagram, SEM images, and X-ray CT images showing the microstructures obtained using different pressing techniques: **a**, **d**, **g** unpressed, **b**, **e**, **h** roll-pressed, and **c**, **f**, **I** isostatic-pressed composite electrodes. The green and black colors indicate pores and occupied AM/SE composite, respectively. The scale bars are 50 μm.

distribution with the quaternary system including electronically conductive agent and binder. However, on average, this model reliably represents the empirical electrode microstructure excluding the relatively negligible electronically conductive agent and binder domain (content of carbon <4.5 vol%). This indicates that those models are representable for the practical SSB electrode microstructure on average, and our electrode model is effective and comparable to the complex digital-twin model and the practical electrode microstructure.

Before the balance threshold, AM-poor cases (Fig. 2a, c) demonstrated lower energy densiy than AM-rich cases (Fig. 2b, d) as shown in Supplementary Fig. 7. This arises from the higher amount of AM integrated into the same CCP-structure in AM-rich cases, thus resulting in higher energy densiy. Meanwhile, AM-poor cases shown improved power performance also exhibiting the higher capacities (177.15 and 190.26 mAh g$^{-1}$) than AM-rich cases (165.79 and 175.83 mAh g$^{-1}$) under the same specific current (36 mA g$^{-1}$). For the percolation threshold, the porosity should be lower than 26 vol% to provide the ionic percolation network in the core-shell CCP model as shown in Fig. 1e. The well-percolated cases (Fig. 2a, b) show the lower porosity translating to higher electrode density and greater capacity. This result indicates that the more efficient ionic transport pathway is provided for well-percolated systems due to the densification. Although the SE network covering AM cannot be percolated in the core-shell model case in the condition where the porosity is over 26 vol%, interestingly, the electrochemical operation is feasible for practical experiments and the polydispersed case even with higher electrode porosity. It is because the polydispersed AM and SE contacted each other, and SE did not

evenly cover AM like the core-shell model in real cases. Given that all the selected model systems represent carbon and binder-deficient electrodes, poorly percolated cases (Fig. 2c, d) show lower capacity than well-percolated cases resulting from poorly connected ionic transport interfaces.

Despite the limitation of the ideal assumption that can be applicable to the case over the percolation threshold, this proposed principle is quite compelling in delivering quantitative design guidelines. Our cross-validation, which incorporates both modeling and experimentation, highlights the robustness of our design principle for high specific energy and high specific power oriented SSBs. Furthermore, this simplified principle can provide clear and rational guidance for electrode design without the need for complex calculations within the condition over the percolation threshold. As a result, it has the potential to create a versatile design framework by combining various design parameters.

## Densification process strategy for enhanced AM/SE interfacial contact with low ASR

Fabricating a composite electrode with low porosity and high interfacial contact over a percolation threshold is important for inducing a low ionic ASR in SSBs. To achieve low porosity and high density in the composite electrode, a specialized compression technique should be developed for practical SSBs[39]. As shown in Fig. 3a–c, the electrode porosity and density vary depending on the densification methodologies. Without any pressing, the interface is primarily disconnected, leading to a highly porous electrode with low density (Fig. 3a), which can lead to high ionic ASR and thus high cell resistance. With conventional roll pressing, a standard method for electrode compaction in

LIBs[47], the interface becomes partially connected, causing decreased porosity and increased density (Fig. 3b). However, it is still not dense enough because the compression is unidirectional. The electrodes swell perpendicular to the pressure to reduce the increasing density. Therefore, to uniformly press the composite electrode and achieve the desirable low porosity and high density, the warm isostatic press (WIP) method was introduced. This process is effective in terms of applying constant pressure to all sides with three-dimensional compression, which is especially efficient for improving interfacial contact with a small amount of SE compared to the AM in the composite electrodes (Fig. 3c). Compared to the other compression techniques, WIP was the only process to reach the percolation threshold. SPEs as the catholyte are usually introduced to the composite electrodes using the infiltration method for conformal AM/SE interfacial contact owing to their flowability. However, this method leads to excessive SE only allowable in porous electrodes, which are unsuitable for high specific energy SSBs. Thus, flexible SPEs should also be added at the AM/SE composite mixing step, which should be followed by consequent isostatic pressing to achieve dense AM-rich, SE-poor electrodes with a uniform interface.

To determine the effect of the compression method on the resulting porosity and density, composite electrodes were prepared using different compression methods and characterized including SEM/EDS, X-ray CT, and FIB (Fig. 3d–i and Supplementary Figs. 8–9 and Supplementary Notes 6–7). Here, the electrode was prepared with a weight ratio of 94:4:2 for NCM622, SPE as a binder, and the electronically conductive agent. Even though Fig. 1 presented the ideal particle and electrode design without consideration of the binder and electronically conductive agent, these components must be added to make the electrode in practice. According to the SEM image, the unpressed electrode contained large voids, whereas the roll-pressed one contained small voids (Fig. 3d, e). Moreover, the electrode prepared using the WIP method contained almost no voids (Fig. 3f). The measured porosity of the WIP-fabricated electrode is almost the same as the calculated value depending on the electrode density in the X-ray CT image in Fig. 3g–i; however, the electrode porosities for the other compressing methods differed substantially. This finding indicates that an appropriate level of compression is necessary to decrease the electrode porosity by reducing the voids and increasing the electrode density and that the WIP method is efficient for fabricating the ideal electrode structure assumed in Fig. 1. The WIP method can lead to low ionic ASR of the composite and, consequently, low overpotential at the cell level. According to Fig. 1h, the unpressed and roll-pressed electrodes are located below the percolation threshold, whereas the electrode prepared using the WIP is positioned above the percolation threshold based on the electrode porosity. Nevertheless, the WIP technique is unlikely to be applied in the continuous manufacturing process because of its single-use waste disposed of pre and post-preparation; therefore, an alternative 3D pressing method should be developed.

## Design strategy of electrode architecture for high-specific-energy SSBs

In terms of electrode design for achieving high specific energy SSBs, it is also critical to determine the ideal loading weight as well as negative electrode combinations. However, there is a lack of publications on how to rationally set these design parameters in previous SSB research. Therefore, we propose rational architecting principles to determine the electrode loading weight and negative electrode match-up. Figure 4a shows the cell configuration for specific power oriented SSBs. Thin electrodes (typically less than 50 μm) were used in this design to attain lower resistance and thus high specific power. In this architecture, the use of the current collector enables a facile electric supply; however, it could constitute a large dummy portion, decreasing the specific energy at the cell level when increasing cell stacks. To decrease

the stack number to increase the specific energy of SSBs, thicker electrodes must be used (Fig. 4b). For example, when the loading weight of the electrodes is doubled, the stack number for producing the same energy is reduced to half. Therefore, the use of the current collector would decrease, leading to a total weight reduction of the cell and, consequently, increasing the specific energy. Moreover, to further reduce the total weight of the cell, the negative electrode can be changed from graphite to Li metal because it has a higher theoretical specific capacity (3860 mAh g$^{-1}$ compared to 372 mAh g$^{-1}$ for graphite) (Fig. 4c). This design strategy is well known[48] and inevitable, especially for SSBs because the composite electrodes are usually heavier than those in LIBs because of the introduction of heavier SEs rather than liquids. In addition, we can also examine other strategies for further increasing the specific energy. To reduce the total weight of the cell, some researchers suggested to adopt a specific cell configuration (termed as "anode-free" or "anode-less") where the negative electrode initially comprises only the current collector with no active material (e.g., lithium)[6]. This design strategy can be quite effective for achieving higher energy density (energy per volume) but is not optimal for increasing the specific energy because of the use of heavy Cu (8.96 g cm$^{-3}$). However, using light Li (0.534 g cm$^{-3}$) instead of Cu may be sufficient to achieve higher specific energy (Fig. 4d). Freestanding thin Li foil (40 μm) can be used as a bifunctional lithiophilic current collector with additional Li inventory. This approach can reduce the weight of the negative electrode to 6 wt% in the same unit volume compared to the Cu-based anode-free configuration. Each design strategy depends on the desired cell characteristics; however, it is critical to control the electrode design parameters rationally.

In terms of the electrode total loading weight described in Fig. 4a, b, there is no intuitive design rule to identify the optimal level; therefore, we propose the use of Ohm's law to determine the electrode loading based on the ionic ASR of the composite building blocks, as shown in Fig. 1b, c. If a certain current rate induces the desired electrode IR drop, we can calculate the electrode thickness through a series of ionic ASR in unit building blocks (Fig. 4e–h). In these figures, for example, we set the desired current rate to 0.5 C, and then electrode IR drops were plotted as a function of the areal loading weight at the AM/SE composition of the balance threshold. For SEs with higher ionic conductivities, the IR drops were less sensitive to the loading weight (Fig. 4e, f). Nevertheless, those with lower ionic conductivities showed dramatic IR drops highly dependent on the loading (Fig. 4h). Because we used SPE as a model SE with an ionic conductivity of about 1 mS cm$^{-1}$ at 25 °C (Fig. 4g), it is possible to point out the electrode loading weight at the 100 mV IR drop borderline of ~21 mg cm$^{-2}$. This is the so-called "loading threshold" equivalent to 73 μm thickness if we assume that the electrode density was ~3 g cm$^{-3}$, indicating the ionic percolation threshold, as shown in Fig. 1e. The detailed calculation process is presented in Supplementary Note 8. The IR drop is underestimated as it only accounts for the positive electrode overpotential and does not include other polarizations. Consequently, it cannot directly correlate to the IR drop at the cell level. In spite of that, this value can provide insights into the design of positive electrode parameters to achieve the desired electrochemical kinetics.

To identify the effect of the negative electrode combination in Fig. 4c, d, the specific energy landscapes at the electrode level were calculated and the contour was plotted in Fig. 4i–l. The detailed calculation process is presented in Supplementary Note 9. Here, NCM622 was selected as a model AM and the electrode specific energy including the Al current collector (20 μm) is presented in Fig. 4i. With increasing areal electrode loading weight and AM ratio, the electrode specific energy also increases until 500 Wh kg$^{-1}$. However, the specific energy might decrease with the addition of more negative electrodes in the system, as shown in Fig. 4b–d. Figure 4j shows the electrode-specific energy with the use of a graphite negative electrode and Cu current collector (10 μm), as depicted in Fig. 4b. However, if the

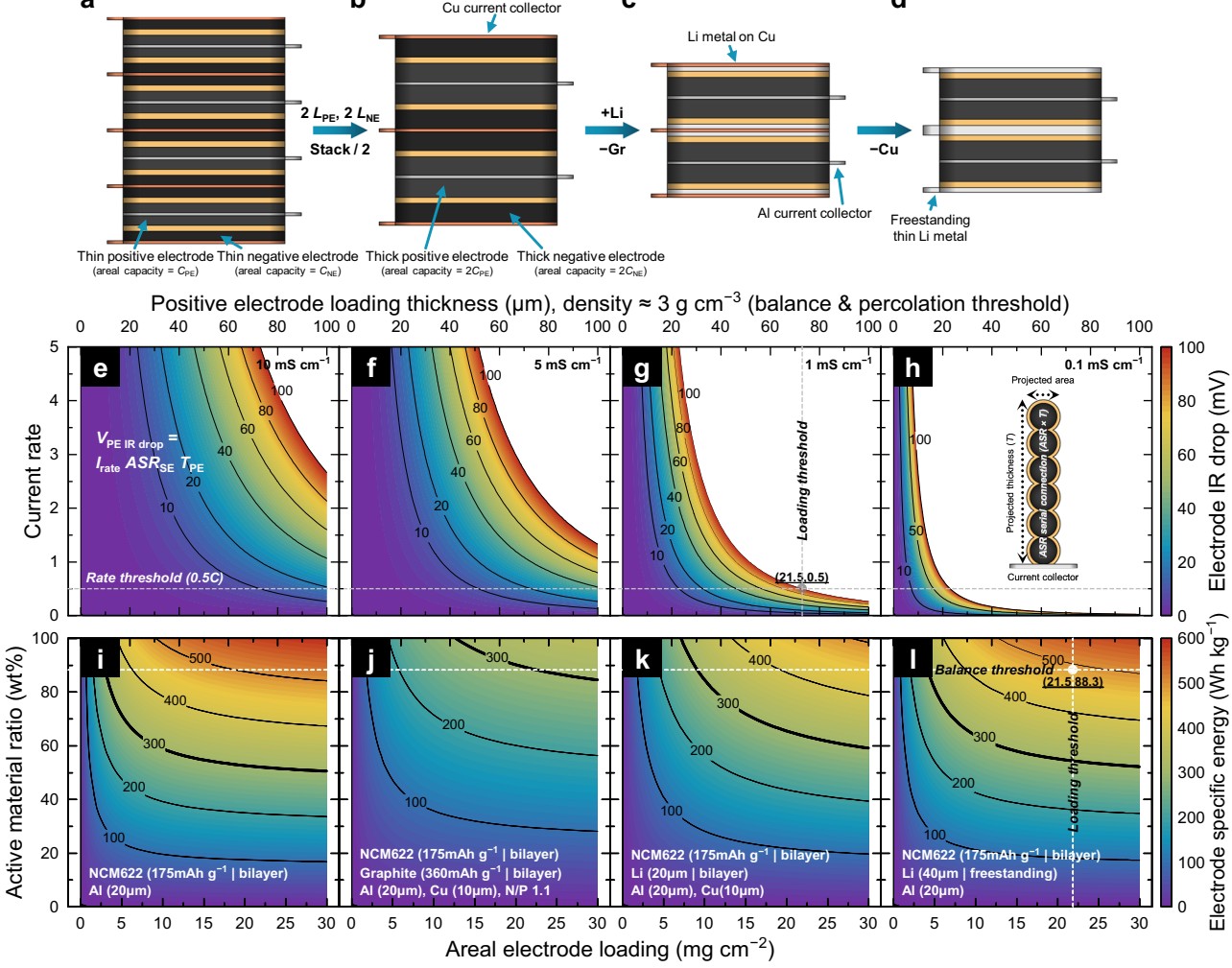

**Fig. 4 | Electrode loading design and combination strategies for high-specific-energy SSBs.** Schematic diagram showing the SSB cells with **a** conventional specific-power oriented design (thin electrode combination). Areal capacity of thin positive electrode is $C_{PE}$ and areal of thin negative electrode capacity is $C_{NE}$ with graphite (Gr), **b** conventional specific energy oriented design (thick electrode combination) when the areal loading weight of the positive and negative electrodes are doubled ($2 L_{PE}$, $2 L_{NE}$, respectively), the areal capacity of thick positive electrode ($2 C_{PE}$) and areal of thick negative electrode capacity ($2 C_{NE}$), **c** advanced specific energy oriented design (thick electrode, thin Li metal negative electrode on Cu current collector), and **d** proposed specific energy oriented design (thick electrode, freestanding thin Li metal negative electrode without current collector). **e**–**h** Electrode IR drop as a function of target current rate (a value of 1 indicates 1 h of charge or discharge for a certain specific capacity, specific capacity in this figure is variable) and areal electrode loading using SEs with ionic conductivities (not obtained by experimental measurements) of **e** 10 mS cm⁻¹, **f** 5 mS cm⁻¹, **g** 1 mS cm⁻¹, and **h** 0.1 mS cm⁻¹. **i**–**l** Electrode specific energy as a function of areal electrode loading and AM ratio with **i** only NCM622 electrode on Al current collector and for the cases with additional **j** graphite negative electrode, **k** thin Li metal negative electrode on Cu current collector, and **l** freestanding thin Li metal negative electrode without current collector.

graphite negative electrode is replaced with Li metal on a Cu current collector (Fig. 4c), the specific energy landscape expands toward a higher level, as shown in Fig. 4k. In addition, if the Cu current collector is removed (Fig. 4d), the design space is further extended, which is almost equivalent to leveraging lightweight freestanding thin Li instead of Cu. For example, at the balance threshold of AM (74 vol%) and areal electrode loading threshold (21 mg cm⁻²), the calculated cell-specific energy is 508 Wh kg⁻¹ with only the NCM622 electrode (Fig. 4i) but is reduced to 298 Wh kg⁻¹ with the graphite negative electrode (Fig. 4j), 413 Wh kg⁻¹ with the thin Li metal negative electrode on the Cu current collector (Fig. 4k), and 486 Wh kg⁻¹ with the freestanding thin Li metal negative electrode (Fig. 4l). The latter is close to the specific energy with only the NCM622 electrode, indicating that the freestanding thin Li metal negative electrode is beneficial to design high specific energy SSBs without gaining almost any weight in the electrode. In Fig. 4l, two baselines can be used to estimate the cell performance, referred to as the balance threshold and loading

threshold and highlighted as white lines. On the right side of the loading threshold, high specific energy can be achieved; however, the specific power will be reduced due to the increased IR drop, as illustrated in Fig. 4e–h. Conversely, on the left side of the loading threshold, high specific power can be achieved with decreased IR drop; however, the specific energy will decrease. This indicates a trade-off relationship between the specific energy and specific power, and the electrode design parameters should be carefully addressed to optimize the cell performance toward the desired direction.

## Cell architecting and assembling strategy for high-specific-energy SSBs

To fabricate a complete unit cell, consideration of the passive parts is also essential. However, the specific energy discussed above was the value considering only the electrodes, not including the SE and other packages. Therefore, we investigated the effect of the SE on the specific energy by developing a fabrication strategy and applying it to

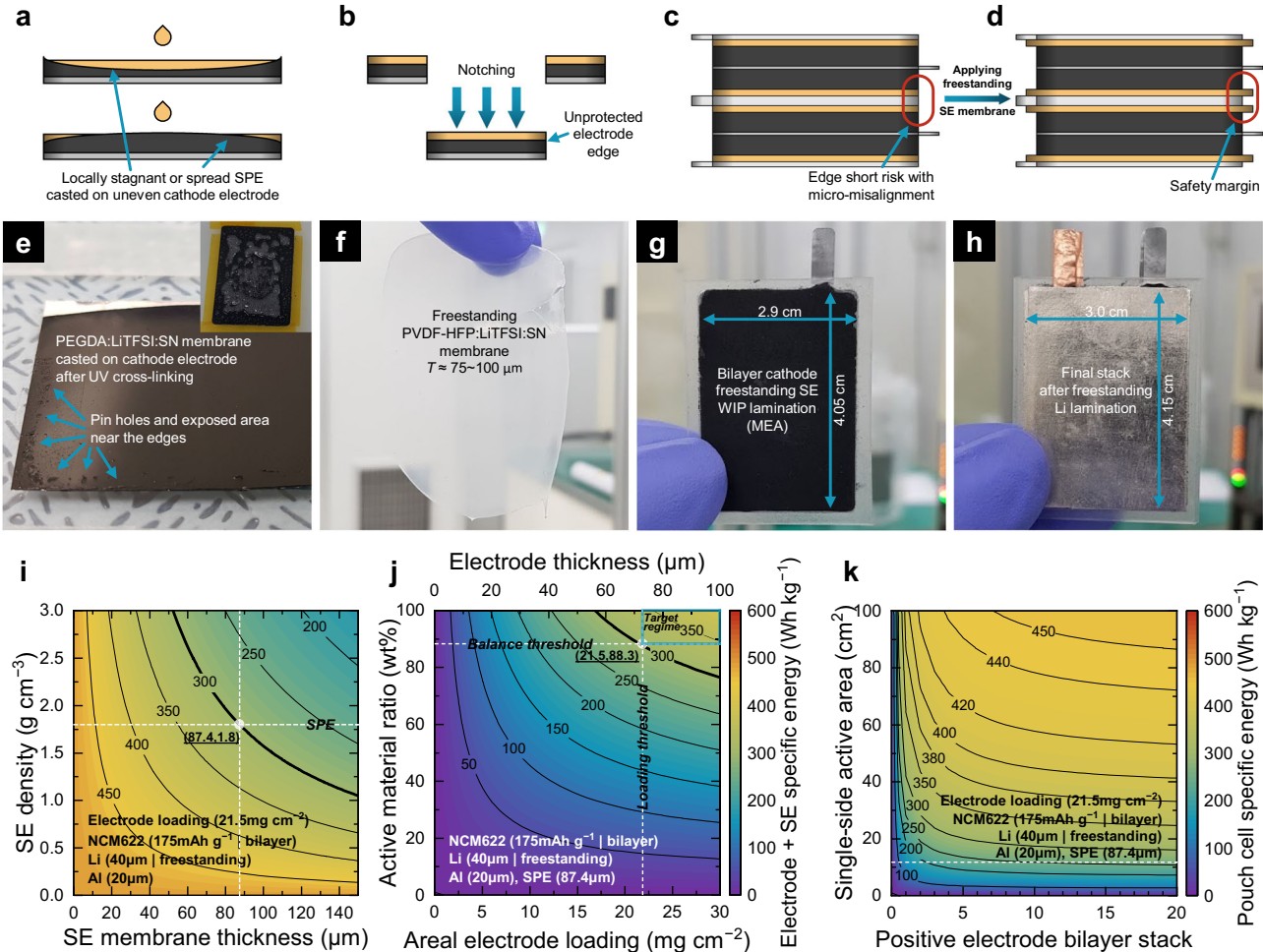

**Fig. 5 | Macro architectural design strategy for reliable high-specific-energy SSBs.** Schematic diagram showing SE membrane fabrication with **a** locally stagnant casted slurry with insufficient amount on unflattened electrode and **b**, **c** unprotected electrode edge after notching and stacking with desirable dimensions. **d** Electrode stacking structure with freestanding safety margin of SE. **e** Photograph of PEGDA–LiTFSI–SN-based SPE membrane cast on electrode after UV cross-linking showing the pin holes and exposed area (inset: vacuum-driven SPE infusion into the electrode pores). **f** Photograph of freestanding PVDF–HFP–LiTFSI–SN-based SPE freestanding membrane (82–96 μm thickness). **g** Bilayer electrode laminated with freestanding SPE membrane (MEA) using WIP. **h** Final jelly roll after freestanding thin Li metal lamination (40 μm thickness). Specific energy normalized by weight of electrode and SE as a function of **i** SE true density and SE membrane thickness, **j** AM ratio, areal electrode loading, and **k** electrode stack number and size based on Fig. 3l.

elaborate the established theoretical electrode design principles presented in Fig. 4. As shown in Fig. 5a, the conventional wet method for casting a thin SE membrane induces a locally stagnant area where the slightly unflattened electrode is exposed without covering by the SE[49]. To solve this problem, the SE slurry should be overpoured on the electrode, which is not suitable for high specific energy SSBs. Furthermore, even if the SE membrane is uniformly coated on the electrode, the unprotected edge might be exposed when the electrode is notched to the desired shape (Fig. 5b). This means the SE membrane as a separator has the same area as the electrode, leading to edge-shorting risk with micro-misalignment (Fig. 5c). Thus, as shown in Fig. 5d, the cell structure with a freestanding SE safety margin can be a solution such as in a typical LIB. With this structure, edge shorting can be prevented, and consequently, the long-term reliable cycling of SSBs can be achieved.

The processing technique and related materials should be changed entirely to build this small edge structure. Photographic pictures of casted and freestanding SE membranes are presented in Fig. 5e–f, respectively. We adopted a well-known poly(ethylene glycol) diacrylate (PEGDA): LiTFSI: SN SPE matrix with UV crosslinking treatment for the SE casting. As seen in Fig. 5e, the casting method induces the formation of tiny pin holes and exposed areas near the edges, leading

to cell shorting after the cell assembly. The pin holes are expected to be formed during the infusion of SE into the electrode pores; thus, the amount of SE should be in excess to form the separator membrane during the SE permeation without tiny defects. Unfortunately, it also means excessive solid-state catholyte (i.e., the solid-state electrolyte contained in the composite positive electrode) will infiltrate electrodes with high porosity. Therefore, this technique would be insufficient for a lean catholyte or less porous electrode. The SE membrane casting and addition of the catholyte should thus be technically separated. In Fig. 5f, we cast a freestanding PVDF-HFP:LiTFSI:SN-based SE membrane. The thickness is tunable; however, we intentionally fabricated a slightly thick membrane (75 - 100 μm) to accommodate the amount of charge from the electrode. To see the effect of the thickness of SE membrane on the cycling stability, the coin cell performance (Li|SPE| NCM622) using different thicknesses of SE membranes was shown in Supplementary Fig. 10 and Supplementary Note 10. This data could prove that the thicker SE membrane is conditionally necessary for operating the cell with high loading positive electrodes. Then, the SE membrane and SPE catholyte-containing composite electrode were laminated to make a membrane–electrode assembly (MEA) using the WIP method, having an additional safety edge margin width of 1 mm (Fig. 5g). Consequently, the freestanding thin Li metal negative

electrode (40 μm) was also gently laminated on the previous stack to fabricate the SSB unit cell (Fig. 5h). The final laminated stack cell had dimensions of approximately 3 cm × 4 cm, which was uniform without any defects.

To evaluate the effect of the SE properties on the plot in Fig. 4l, the cell-specific energy was plotted as a function of thickness and true density, as shown in Fig. 5i. The detailed calculation process is discussed in Supplementary Note 11. This plot shows that the thickness of SE membrane should be about 87 μm to reach the specific energy of 300 Wh kg$^{-1}$ if SPE with 1.8 g cm$^{-3}$ of true density was used and normalized by the mass of electrodes and SE membrane. It is reasonable for the specific energy to decrease as these parameters increase. These SE characteristics are often underestimated in previous literature; however, they should be carefully addressed with proper processing techniques toward higher specific energy SSBs. If we use the same design space as that in Fig. 4l with the addition of SE weight marked at the intersecting point of the white dashed lines in Fig. 5i, the entire specific energy equivalently decreases, as shown in Fig. 5j. Thus, according to this design strategy, the target regime can be derived from the marked balance and loading thresholds. This foundation can also be adapted to evaluate the electrode size and stack numbers while adding the aluminum pouch and lead tab weights (Fig. 5k). It is well known that upscaling the cell size efficiently boosts the specific energy by diluting the portion of passive parts in the case. As seen in Fig. 5k, the electrode size and stack number are both critical to the specific energy; however, those should be scaled up to enter the practically useful range of the high-level regime[42]. According to our design principles, over 12 stacks of positive electrode bilayer are required to achieve a pouch cell specific energy of more than 250 Wh kg$^{-1}$ given the cell size fabricated (11.75 cm$^2$).

## Electrochemical characterizations of high-specific-energy SSB pouch cell based on SPE

Using the described design principles above, we demonstrated practical SSB pouch cells with high specific energy, which is relevant to commercially available LIBs. The electrode design was tuned toward the specific energy orientation with higher AM and lean SE contents as well as higher electrode loading to reach the target regime guided by the balance/percolation/loading thresholds in Fig. 5j. The design sheets of these parameters are presented in Supplementary Tables 1–3 for the pouch cells with specific energies over 203, 281 and 310 Wh kg$^{-1}$, respectively. Notably, the described specific energies were normalized by the total weight of all the parts including the electrodes, current collectors, SE membranes, lead tabs, and packaging materials.

Supplementary Fig. 11a presents the initial electrochemical charge/discharge profile for the pouch cell with a specific energy over 200 Wh kg$^{-1}$. The composite electrode was composed of 93 wt% AM and 5 wt% SE catholyte. The loading weight was ~19 mg cm$^{-2}$, equivalent to an areal discharge capacity of 3.32 mAh cm$^{-2}$ and a specific capacity of 187 mAh g$_{AM}^{-1}$ with NCM622. Using average 82 μm-thick SE membranes, two double-sided positive and three negative electrodes were alternately stacked to build the pouch cell with a total discharge capacity of 156 mAh, and the nominal discharge voltage was ~3.81 V at the testing temperature of 45 °C. The coulombic efficiency was over 90%, and the energy efficiency was ~88% due to the difference in the nominal charge/discharge voltages. For this design, the entire pouch cell weighed ~2.92 g, indicating a specific energy of ~200 Wh kg$^{-1}$. The gravimetric composition of the pouch cell is depicted in Supplementary Fig. 11b. Even if the embodied specific energy was reasonably good with the advanced composite electrode design, the AM portion, which is critical to the specific energy, was only ~28 wt%. The other passive components sacrificed the specific energy. This is because of the limited cell capacity linked to the stack numbers, which is insufficient to minimize the other components. Therefore, the cell architecting and

processing methodologies, including the electrode stacking and size scale-up, are critical for high specific energy design.

To further increase the specific energy toward the practical level, as shown in Fig. 6a, the electrode design was slightly tuned to the specific energy orientation, containing 94 wt% AM while maintaining the SE catholyte at 4 wt%. The total loading weight was increased to over 26 mg cm$^{-2}$, equivalent to an areal discharge capacity of 4.27 mAh cm$^{-2}$ and a specific capacity of 173 mAh g$_{AM}^{-1}$ with NCM622. The SE membrane thickness also was increased to ~98 μm on average to account for the enhanced charge utilization. Ten positive and eleven negative electrodes with double-side coat were sequentially laminated to build a total discharge capacity of 1 Ah, and the nominal discharge voltage was slightly decreased to ~3.75 V. The coulombic efficiency was >87% and the energy efficiency was ~83% at the first discharge because of the increased overpotential from the thicker electrodes and SE membrane. The weight of the entire pouch cell was 5.79 g, indicating a specific energy of ~280 Wh kg$^{-1}$. This specific energy was also identical to the energy density of ~600 Wh L$^{-1}$, even when including the whole package. The specific energy from the second cycle was certified by an external third-party organization (Supplementary Fig. 12). The composition of the entire parts was also calculated, as shown in Fig. 6b. Compared to Supplementary Fig. 11b, the fraction of the AM was boosted to almost 43 wt%, whereas that of the other inactive parts was decreased. This is because of the scaled-up electrode size and stacks of the SSB pouch cell, and the importance of prototyping toward practicality is highly significant. Furthermore, the highest specific energy of 310 Wh kg$^{-1}$ of SSB pouch cell could be obtained with the higher total loading weight of 30 mg cm$^{-2}$, equivalent to an areal discharge capacity of 5.24 mAh cm$^{-2}$, as shown in Fig. 6c, d. In summary, the achieved specific energy in the SSB pouch cell demonstrations (200, 280, and 310 Wh kg$^{-1}$) are similar to the estimated specific energy from the designed parameters (203, 281, and 310 Wh kg$^{-1}$), indicating that this design guideline is valid for producing the desired specific energy of SSBs.

However, some battery performances, such as long-term cyclability, are not above the state-of-the-art. The reasons are varied. As shown in the Li‖NCM622 coin cell tests in Supplementary Fig. 13 and Supplementary Note 12, the cell electrochemical properties were improved by tuning the electrode loading parameters. Here, NCM622 was chosen as a AM material instead of NCM811 even though the specific capacity is lower because of better chemical stability (Supplementary Figs. 14–15 and Supplementary Note 13). Furthermore, the galvanostatic intermittent titration technique (GITT) result of coin-cell was shown in Supplementary Fig. 16 to confirm the comparable diffusivity (~10$^{-9}$ cm$^2$ S$^{-1}$ at 25 °C) in our system compared with the general LIB system. We would like to point out that these diffusivity values are aligned with the state-of-the-art[50–52], and it means there is no problem with AM caused by the discrepancy of diffusivity with the LIB system. XPS measurements of the Li metal negative electrode and FTIR measurements of SPE after the cycling test were analyzed, and the results indicated that stable LiF and Li$_3$N were formed with almost no structural change in SPE during cycling (Supplementary Figs. 17, 18 and Supplementary Notes 14, 15). However, this performance is still not comparable to that of LIBs, potentially because of a possible chemical side reaction, such as the dehydrofluorination for PVDF-based polymers[53] (Supplementary Fig. 3) and the polymerization of nitrile groups in SN catalyzed by Li metal, incurring high interfacial resistance[54–57], and increased resistance because of high positive electrode loading weight and a large number of electrodes in the pouch-type stack. Supplementary Fig. 19 shows several methods, such as washing, annealing the active material, and lowering the operation temperature, which was tried to suppress the dehydrofluorination triggered by the residual lithium on the active material surface and elevated temperature. The electrochemical performances from all tries are still suddenly dropped after approximately 150~160 h. However, given that the cell shows an improved cycle number (10 cycles) by

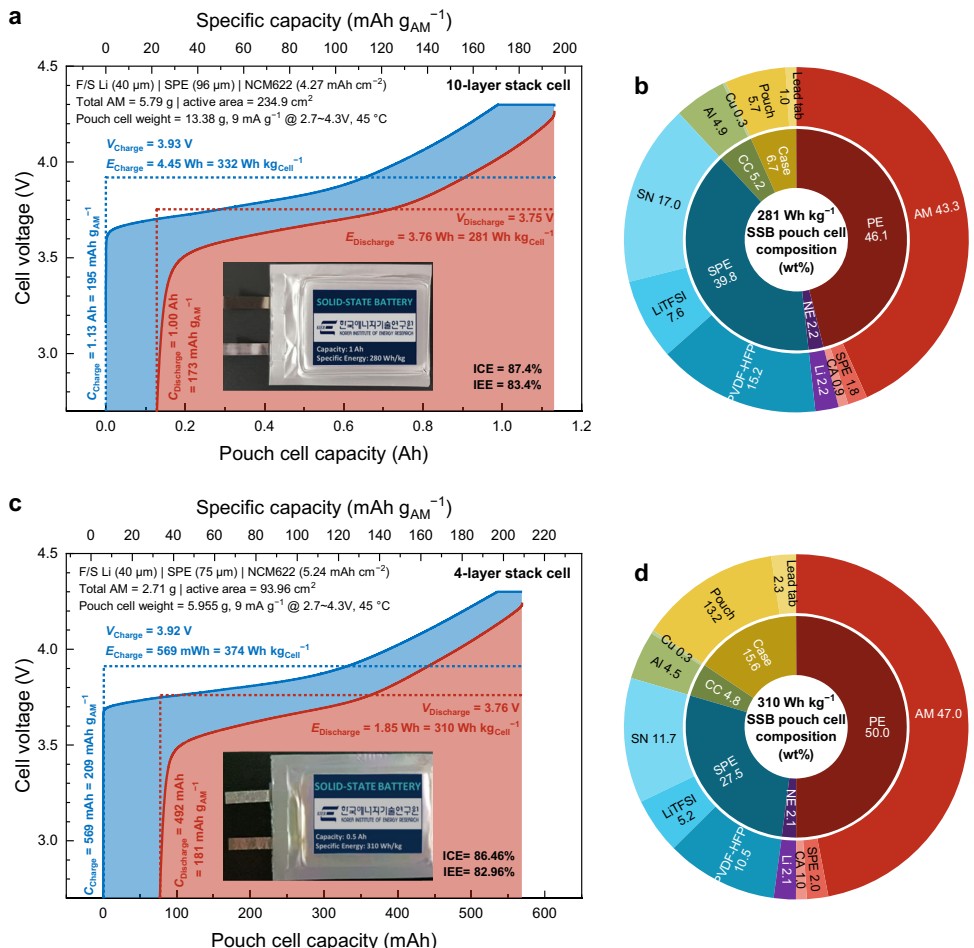

**Fig. 6 | Electrochemical characteristics and compositions of practical SSB pouch cells.** Initial charge–discharge electrochemical profiles and weight compositions of all the configuring materials of SSB pouch cells with **a**, **b** 1 Ah scale, 280 Wh kg⁻¹, and **c**, **d** 0.5 Ah scale, 310 Wh kg⁻¹. The 1 Ah and 0.5 Ah-scale cell are composed of four and ten bilayer electrodes, respectively, and three and eleven freestanding thin Li metal negative electrodes, respectively. The testing was conducted within operating cell voltage range of 2.7–4.3 V, and the condition was set to 9 mA g⁻¹ under 45 °C. The composition included all the inactive components of the cell, including the case.

controlling the charge/discharge protocol, our design logic is a proper guideline for designing the electrode. We want to point out that the SPE strategy we propose here is a proof-of-concept which require future work by the research community. Even so, we expect this to be a matter of unoptimized chemistries rather than cell design engineering.

However, the demonstrated high specific energy over 300 Wh kg⁻¹ of the SSB pouch cells are still noteworthy. Even though numerous corporations have claimed to develop high specific energy SSBs with over 300 Wh kg⁻¹, public access to detailed information for peer verification is prohibited. However, for a fair comparison to evaluate this accomplishment, we collected related information from the literature of Janek et al.[5] and several representative research articles and then quantitatively assessed them using the provided model function, including this data in Fig. 7 and Supplementary Table 4[13–31,36,58–66]. The bubble plot indicates the areal electrode loading weight (mg cm⁻²), AM ratio (wt%), areal discharge capacity (mAh cm⁻²), and the specific energy normalized by the weight of the electrode and SE membrane. We should note that the specific energies in Fig. 6 and Fig. 7 are different. Fig. 6 (310 Wh kg⁻¹) incorporated all the weights of the pouch cell, including even the aluminum pouch and lead tab. The latter in Fig. 7 (433 Wh kg⁻¹) included only the electrodes and SE weight for possible comparison with the other studies. Although their testing conditions differ slightly in terms of the current rate and temperatures, the basic information about the electrode design parameters and the derived specific energies can be

compared in the same design space. The figure shows that most previous studies mark lower electrode loading and AM ratio, which is insufficient to achieve high specific energies. However, several previous works rank higher in terms of electrode loading and AM ratio at the electrode level, resulting in improved areal discharge capacities. Still, a few only demonstrated the cell-level performance with proper prototyping; therefore, the specific energy characteristics of this study are considerable on their own in terms of the advanced electrode design as well as the practical cell demonstration achieving 310 Wh kg⁻¹ at the 0.5 Ah scale.

To summarize design principles for high specific energy SSB electrodes and experimental verification with multi-stacked pouch cells were introduced in this study. The design logic covers multiscale and multi-parameters from the particle microstructures to the practical cell-based macro architecture. In these guidelines, the various design spaces were provided, and three design thresholds essential at the electrode level were developed and provided to rationalize the SSB design strategy: 1. the balance threshold to optimize the AM/SE ratio in the electrode composition based on the CCP structure; 2. the percolation threshold to achieve an effective electrode density to ensure AM/SE interface contact, which is also based on the CCP arrangement; and 3. the loading threshold, which determines the optimal loading weight for the target current rate and IR drop based on Ohm's law and the known ASR. We developed the design principles and prototyped a high specific energy SSB pouch cell with a specific energy over 280 and

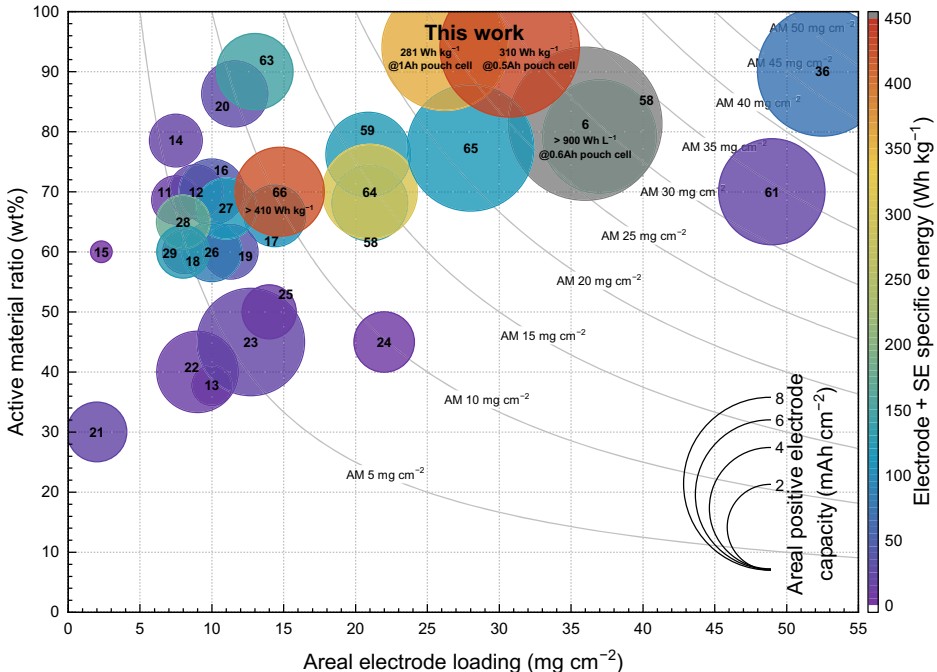

**Fig. 7 | Performance benchmarking of SSBs against recently published studies.** For fair comparison, the specific energy was normalized by the weight of the electrodes and SE membrane using the calculation provided by Janek et al.[5]. The horizontal axis represents the areal electrode loading (mg cm⁻²), and the vertical axis indicates the AM ratio (wt%) of the electrode. The size of the symbol indicates the areal discharge capacity (mAh cm⁻²) of the electrode, and the color represents the specific energy (Wh kg⁻¹). The rational function lines connect the equivalent AM loading amounts among the different electrode designs. The references are presented in Supplementary Table 4.

300 Wh kg⁻¹ at the 1 Ah and 0.5 Ah level using proper processing techniques to demonstrate that the design logic is indeed compelling.

## Methods

### Preparation of electrodes using different pressing processes

A slurry mixture of 94 wt% $LiNi_{0.6}Co_{0.2}Mn_{0.2}O_2$ (NCM622, LG Chem), 2 wt% conductive carbon black (Super P), and 4 wt% binder solution. The binder solution was composed of poly(vinylidene fluoride-co-hexafluoropropylene) (PVDF-HFP, pellets, Sigma-Aldrich) as the polymer host, succinonitrile (SN, > 99%, Sigma-Aldrich) as a plasticizer, and lithium bis(trifluoromethanesulfonic)imide (LiTFSI, > 98%, Sigma-Aldrich) as a Li salt dissolved in N-methyl-2-pyrrolidinone (NMP, > 99.5%, Sigma-Aldrich) as a solvent with a Thinky mixer (ARE-310) at 2000 rpm for 5 min) at 25 °C under 1 atm. This slurry was coated on the Al foil (99.3% purity, 20 μm thickness) as a current collector with a doctor blade at 25 °C under 1 atm and dried at 80 °C for 12 h under vacuum. Three types of samples were prepared: an electrode without any pressing, an electrode after roll pressing, which applies the pressure two-dimensionally, and an electrode after warm isostatic pressing (WIP). The WIP process was divided into three steps. The pressure was slightly increased up to 5500 bar over 5 min during the first step, maintained at 5500 bar over 10 min during the second step, and slightly decreased to zero over 5 min in the last step. The entire process was performed at 70 °C. The electrode thickness of each pressing method is 104, 83.3, and 70 μm corresponding loading level is 25 mg cm⁻². Li metal negative electrode (40 μm, 3.1 cm × 4.25 cm, >99.9%, NEBA Corporation) is sealed under vacuum and stored in a dry room with a dew point −60 °C.

### Characteristic analysis of electrodes prepared using different pressing processes

The morphology and uniformity of the prepared electrode samples were examined using field-emission scanning electron microscope/energy-dispersive X-ray spectroscopy (FE-SEM/EDS, JEOL, JSM-7610). In addition, the focused ion beam (FIB, Helios, G4 UC, Thermo Fisher Scientific) was used to examine the three-dimensional structure of the electrodes. X-ray computed tomography (X-ray CT) was also used to analyze the differences in the porosity and density of the electrodes for the different pressing processes under 1 atm. Finally, for X-ray CT image, the Dragonfly program was used to analyze and calculate the porosity and density of the electrodes. The as-prepared electrodes and electrolytes for ex-situ characterizations were collected in a dry room with a dew point −60 °C, sealed to prevent the degradation of the sample, and then quickly transferred to the equipment after disassembly of the sealed sample except for the X-ray CT analysis.

### Preparation of solid polymer electrolyte

The solid polymer electrolyte (SPE) was prepared using a mixture of PVDF−HFP as the polymer host, SN as the plasticizer, LiTFSI as the Li salt dissolved in acetone as a solvent. The molar ratio of SN and LiTFSI was 8:1, and the weight ratio of LiTFSI and PVDF−HFP was 1:2. SN and LiTFSI were mixed and stirred at 80 °C overnight, and afterward, the PVDF−HFP and acetone were added, and the mixture was stirred at 80 °C overnight. For SE membrane, the prepared SPE solution was casted on a glass plate with a doctor blade (900 μm gap) and dried at 60 °C for 1 h.

### Characteristic analysis of SPE

To measure the mechanical strength of the prepared SPE, dynamic mechanical analysis (DMA, TA, Q800) was performed. Thermogravimetric analysis (TGA) was conducted to identify the thermal stability of the SPE. To analyze the chemical structure of the SPE, Fourier-transform infrared spectroscopy (FT-IR, Mattson, Satelite5000) was used. SPE was prepared in a dry room with a dew point −60 °C and sealed under the vacuum before characterization.

## Electrochemical analysis of SPE and electrode

Electrochemical analysis was performed to determine the electrochemical properties of the SPE and redox properties of the NCM622 electrode. Electrochemical impedance spectroscopy (EIS) analysis of the SS|SPE|SS cell was performed in the frequency range of 100–7 MHz at 25 °C to determine the ionic conductivity of the SPE. In addition, EIS spectra of the SS|SPE|SS cell were collected at various temperatures (30 °C, 40 °C, and 80 °C) to calculate the activation energy ($E_a$) of the SPE. The $E_a$ can be calculated by the Arrhenius equation (Eq. (1)). The lithium-ion transference number was derived using the Bruce-Vincent method based on chronoamperometry (CA) and EIS spectra measured before and after CA measurement of the Li|SPE|Li cell. The lithium-ion transference number can be calculated by Eq. (2). Linear sweep voltammetry (LSV) of the SS|SPE|Li cell was performed to determine the electrochemical stable voltage window. To evaluate the stability of the SPE against a Li metal negative electrode, lithium symmetric cycling tests of the Li|SPE|Li cell were performed with a current density of ±0.15 mA cm$^{-2}$ for 1 h as each cycle. Furthermore, cyclic voltammetry (CV) of the Li|SPE|NCM622 cell was performed to analyze the redox property of NCM622. All the measurements were performed using Biologic VSP system and EC-Lab software.

$$\sigma T = \sigma_0 \exp\left(-\frac{E_a}{RT}\right) \tag{1}$$

(Here, $\sigma$ is the ionic conductivity of the sample, $T$ is the temperature of conductivity measurement, $R$ is the gas constant, and $E_a$ is the activation energy. The slope of the Arrhenius plots changed to a lesser extent depending upon the composition of the solid electrolyte and temperature; hence the activation energy is directly proportional to the slope.)

$$t_{Li^+} = \frac{I_{SS}(\Delta V - I_0 R_0)}{I_0(\Delta V - I_{SS} R_{SS})} \tag{2}$$

(Here, $t_{Li^+}$ is the lithium transference number, $I_0$ and $I_{SS}$ are the initial and steady-state currents, $\Delta V$ is the potential applied across the cell (here, this value was 100 mV.), and $R_0$ and $R_{SS}$ are the initial and steady-state charge-transfer resistances SE membrane.)

## Li|SPE|NCM622 coin-cell test

Li|SPE|NCM622 coin-cell (CR2032) made of SUS 316 L, height and diameter are 20 and 32 mm, tests were performed using different areal electrode loadings of single-side coating. One test was performed with a low areal electrode loading of 3.50 mg cm$^{-2}$, and another test was performed with a high areal electrode loading of 22.67 mg cm$^{-2}$. The prepared SPE was used as the solid electrolyte punched with a radius 9.5 mm. The positive electrodes and SPE were prepared using the WIP. The first step is applying the WIP for each positive electrode and SPE separately. After that, the second step is applying the WIP for membrane-electrode assembly (MEA) which is laminated the positive electrode and SPE for maximizing MEA interfacial contact. The detailed explanation and figures can be seen in Supplementary Fig. 20 and Supplementary Note 16. Li foil was used as the negative electrode for all tests. The coin cells were prepared in a dry room with a dew point −60 °C, and the tests were run at 18 mA g$^{-1}$ in the cut-off voltage range of 3.0–4.3 V at 25 °C for 20 cycles using a potentiostat (WonA-Tech, WBCS3000). The specific current and the specific capacity were calculated by normalizing the value with the mass of positive electrode active materials. To analyze the stability of the Li metal negative electrode and SPE during cycling, XPS (NEXSA) and FT-IR analysis were performed, respectively. GITT measurement was performed with a galvanostatic charge/discharge of 9 mA g$^{-1}$ for 20 min, followed by an open circuit for 1 h. The data point acquired per 10 sec and the slope of potential-$\sqrt{t}$ during electrochemical experiment is averagely 0.015, −0.04 for charge, discharge.

## Li|SPE|NCM622 pouch cell assembly and test

Li|SPE|NCM622 pouch cell tests were performed using different active material weight ratios, areal electrode loadings, number of electrode bilayer stacks, and single-side active areas. One pouch cell used 93 wt% active material and an areal electrode loading of 19.09 mg cm$^{-2}$. A single-side active area of 12 cm$^2$ and a 2-electrode bilayer stack were used. SPE with a true density of 1.8 g cm$^{-3}$ and thickness of 82 μm and Li foil with a thickness of 40 μm were used as the solid electrolyte and negative electrode, respectively. The other pouch cells with specific energies of 280 and 310 Wh kg$^{-1}$ used 94 wt% active material and an areal electrode loading of 26.23 and 30.74 mg cm$^{-2}$, respectively. A single-side active area was 12 cm$^2$, and 10- and 4-double-side coated positive electrodes were stacked. The 40 μm thick Li metal and 96 and 84 μm thick SPE were used as the negative electrode and the solid electrolyte membrane, respectively. Here, the sizes for the negative electrode, positive electrode, and SPE were different: Negative electrode is larger than the positive electrode because the negative electrode should take lithium ions from the positive electrode during the charging. SPE is larger than the electrodes to prevent edge shortening (the schematic figure can be seen in Supplementary Fig. 21). The electrodes were cut by electrode punching machines with the designed sizes for negative and positive electrodes. SPE was cut by a cutter. The MEA (membrane-electrode assembly: laminated positive electrode-SPE) was made by WIP which is the same fabrication process in the coin cell test above for interfacial contact. After that, the electrode tabs and the lead tab terminals were welded. Al pouch film was molded into the electrode shape and the depth was adjusted by the stopper with different heights depending on the number of stacks. To seal Al pouch film, the tab and side were sealed with dedicated machines. A vacuum sealing was introduced to the pouch cell as a final assembly, followed by hot-pressing to enhance the layer contact inside the pouch cell by applying constant heat and pressure. The pouch cell tests were performed in an oven fixed at 45 °C under 18 mA g$^{-1}$ in the cut-off voltage range of 2.7–4.3 V using a Biologic VSP system. The external pressure applied during cycling is 3.74 MPa. The specific current and the specific capacity were calculated by normalizing the value with the mass of positive electrode active materials. The specific energy was calculated by normalizing the discharge energy with the mass of total mass of cell including the packaging case.

## Modeling of 3D composite positive electrode structures

NCM622, SPE, and Super P were used in varying weight ratios (NCM622: SPE: Super P = 94.0: 4.0: 2.0 and 78.5: 19.5: 2.0 wt%) to create a virtual composite positive electrode. Particle size distribution of spherical NCM622 through particle size analysis (PSA, Malvern Panalytical, Mastersizer 3000) was applied for modeling and the monodispersed CCP structure was generated using a single particle size (8.55 μm). NCM622 was covered by SPE as a core-shell structure, following the suggested design principle. Each virtual 3D structure had a voxel length of 0.2 μm and was modeled under design conditions with domain sizes, thickness and electrode densities as shown in Supplementary Table 5. All these processes were conducted using the GrainGeo and ProcessGeo modules of the GeoDict 2023 software (Math2Market, Germany).

## Charge/discharge simulation of 3D composite positive electrode structures

An electrochemical simulation was performed using the Left Identity Right (LIR) solver after composing a digital-twin solid-state cell through the Design battery in the BatteryDict module. The pouch cell is operated in the same condition as the pouch cell test. Each cell in four regions was charged at 18 mA g$^{-1}$ and discharged at 36 mA g$^{-1}$ at 45 °C in the cut-off voltage range of 2.7–4.3 V shown in Fig. 2. The input parameters for this simulation are detailed in Supplementary Table 6.

## Reporting summary

Further information on research design is available in the Nature Portfolio Reporting Summary linked to this article.

## Data availability

The data supporting the findings of this study are available in the paper and its Supplementary Information; further data are available from the corresponding author on request. The raw data are protected and are not available due to data privacy laws. The processed data are available at Manuscript and Supplementary Information. The data generated in this study are provided in the Manuscript/Supplementary Information. SolidXCell for SSB cell design toolkit is available as Supplementary Data 1.

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

## Acknowledgements

This research was supported by a National Research Council of Science & Technology (NST) grant by the Korea government (MSIT) (No. CAP 21043-000, No. GTL24011-000) and by a National Research Foundation of Korea (NRF) grant funded by the Korea government (MSIT) (No. 2022R1C1C1006575 and 2023R1A2C2008242).

## Author contributions

Wonmi Lee, Juho Lee, Taegyun Yu, Sung-Kyun Jung, and Jinsoo Kim are responsible for the experimental design and manuscript preparation. Wonmi Lee, Juho Lee, Taegyun Yu, Min Kyung Kim, Sungbin Jang, and Juhee Kim fabricated the SPE-based pouch cell. Hyeong-Jong Kim simulated the 3D digital-twin model. Yu-Jin Han performed the X-ray CT analysis, and Sunghun Choi conducted the TGA analysis. Sinho Choi, Tae-Hee Kim, Sang-Hoon Park, Wooyoung Jin, Gyujin Song and Dong-Hwa Seo participated in the design logic discussion. Sung-Kyun Jung and Jinsoo Kim conceived the high specific energy SSB pouch-cell design logic and contributed to the manuscript preparation.

## Competing interests

The authors declare no competing interests.
