## [Peer Review File · Nature Communications]

REVIEWER COMMENTS

Reviewer #1 (Remarks to the Author):

The paper describes the process of designing a 1Ah polymer-based solid-state battery. The design process is mainly focusing on geometrical parameters.

The methodology is sound, and it would set a very good example for the scientific community.

The results presented are generally interesting, although not groundbreaking from a performance point of view. While not solid-state, there have been several recent examples of ultra-high energy density batteries (e.g., most recent 10.1088/0256-307X/40/4/048201).

There are some minor concerns with methodologies, which are generally academic standards (e.g., the use of stainless steel for the anodic stability; use of very low current for Li/Li plating tests; these are hardly representative of the real conditions observed during cycling). It would be useful to observe, for example, the critical current density for this SPE.

Other very minor concerns are related to a generally unclear description of the fabrication of electrodes, e.g., if those are simply cut, how these highly defective layers would behave in case of welding procedures, or, more generally, what kind of quality control is used to make the small 10 layer pouch (rejects, etc., for each part).

The main concern is that the paper is limited, focusing only on geometrical parameters. This can significantly underestimate the material required to fabricate a real cell or the chemical nature of said materials. A proper analysis should consider electrochemical variables and other types of resistances to guide material selection and cell design. For example, other types of resistance (e.g., diffusion coefficients, charge transfer) and measured variables (electrode tortuosity, coulombic efficiency, more in-depth description of N/P ratio). It is also unclear why the authors selected a SN-based membrane, given the poor cathodic stability - it is unclear what happens after decomposition.

Considering there is more than one commercial software available for cell design and simulation, this approach is not competitive. The authors should consider expanding the scope for publishing in a journal like Nature Communications.

The results show a relatively poor cycle life, including for the 1 Ah pouch (first cycle being ca. 280 Wh/kg, second one from the third-party being about 265 Wh/kg), an aspect that is not sufficiently considered. To prove their point (that their method is effective) extended cycling over at least a hundred of cycles should be demonstrated.

Finally, given that their article would be open-access, the authors should consider sharing a spreadsheet or file to facilitate the improvement of their cell design model.

Reviewer #2 (Remarks to the Author):

This article presented rational design principles for achieving high energy density in practical solid-state lithium batteries. The author demonstrated the feasibility of this principle through the fabrication of a 10-layer stacked solid-state pouch cell with a solid polymer electrolyte, resulting in an energy density of 280 Wh kg⁻¹ and 600 Wh L⁻¹. Authors need to address the following comments before this paper can be considered further.

1. Page 6 line 15-19: the ionic ASR inside of AM particles (during 0~5μm of core-compressed surface distance) showed different trends between fig1 b and fig 1 c, please give explanations.
2. Page 6 line 6-8: “the threshold is based on an ideal assumption that does not consider the actual particle morphology and size distribution”, in fact, the size distribution and mechanical property of active cathode particles can affect the dense packing state of cathode plates and thus the battery energy-density. The author needs to take these factors into account in the model and modify it to make it more close to the actual situation.
3. Why does the electrolyte thickness reach nearly 100μm? Can the thickness of SE be reduced to further improve the energy density? In previous works, the thickness of electrolytes have been reduced to below 20μm. The thick SE selected in this paper is not representative of high- energy-density batteries.

[1] Toluene Tolerated $\text{Li}_{9.88}\text{GeP}_{1.96}\text{Sb}_{0.04}\text{S}_{11.88}\text{Cl}_{0.12}$ Solid Electrolyte toward Ultrathin Membranes for All-Solid-State Lithium Batteries. *Nano Lett.* 2023, 23, 1, 227–234.

[2] An Ultra-Thin Crosslinked Carbonate Ester Electrolyte for 24 V Bipolar Lithium-Metal Batteries. *J. Electrochem. Soc.* 2022, 169, 090509.

[3] Scalable, Ultrathin, and High-Temperature-Resistant Solid Polymer Electrolytes for Energy-Dense Lithium Metal Batteries. *Adv. Energy Mater.* 2022, 12, 2103720.

4. The purpose of this paper is to prepare high-energy-density pouch cells, why not choose materials with higher energy density (such as NCM811) as the cathode active material?

Reviewer #3 (Remarks to the Author):

The manuscript is focused on the engineering, and the WIP technique is not applicable in the cathode manufacturing process. Additionally, some key information is missing, also there are too many mistakes in manuscript.

Thus, I am not in favor of publishing this article in *Nature Communications*. The quality is not close to the standard of this journal.

1. The transference number of the polymer electrolyte is 0.9, which means that the transportation of anions is selectively restricted. And this value is higher than that of most reported works. The authors had better provide a more detailed explanation, maybe NMR experiment is needed. Additionally, the method to EIS fitting and the calculation method to the transference number must be provided.

2. Some key experimental information and methods are missing. For example, test and calculation methods of activation energy, preparation of freestanding thin Li-metal anode, and so on.

3. The current density used in the plating-stripping tests of the $\text{Li}|\text{Li}$ cell was too low (0.15 mA cm^{-2}). The current should reach at least 2 mA cm^{-2} for practical applications.

4. Page 8, line 1: resistance would thus be minimized using this structure. The author had better give the resistance values with or without using this structure.

5. Page 11, line 5: X-ray CT images are shown in Figure 2g-h, not Figure 1h.

6. Page 12, line 24: I don't understand the description of Fig. 1b, c. It Maybe Supplementary Fig. 7b, c?

7. The cycling performance of the cell is not enough.

Response Letter

Manuscript NCOMMS-23-18937-T

Title: Ultimate design principles and fabrication process for high-energy-density solid-state pouch cells based on solid polymer electrolytes

Authors: Wonmi Lee, Juho Lee, Taegyun Yu, Hyeong-Jong Kim, Min Kyung Kim, Sungbin Jang, Juhee Kim, Yujin Han, Sunghun Choi, Sinho Choi, Taehee Kim, Sang-Hoon Park, Wooyoung Jin, Gyu-jin Song, Dong-Hwa Seo, Sung-Kyun Jung, and Jinsoo Kim

Reviewer #1

The paper describes the process of designing a 1Ah polymer-based solid-state battery. The design process is mainly focusing on geometrical parameters. The methodology is sound, and it would set a very good example for the scientific community.

→ **Response:** We are thankful for the reviewer's positive assessment of our work and agree with the reviewer that "*The design process is mainly focusing on geometrical parameters. The methodology is sound, and it would set a very good example for the scientific community.*" in order to spur the systematic design of various parameters for pursuing high energy density solid-state batteries. With the revision of our manuscript in the following, we hope that our conclusion has been further reinforced.

Continued from the comment of Reviwer #1:

1-1. The results presented are generally interesting, although not groundbreaking from a performance point of view. While not solid-state, there have been several recent examples of ultra-high energy density batteries (e.g., most recent 10.1088/0256-307X/40/4/048201²).

→ **Response:** Thank you for your comment. From a performance point of view, cycling stability and energy density still should be improved further in future work by developing optimized materials. The reference paper that you mentioned in the comment showed the high energy density (711.3 Wh kg⁻¹)

of pouch-type LIBs using a high-capacity cathode (LNCMO) and lithium metal anode under practical conditions (high one-sided area capacity of the cathode of $> 10 \text{ mAh cm}^{-2}$, compact density of the cathode of 2.6 g cm^{-3} , E/C ratio of 1.3 g Ah^{-1})². It is quite impressive, but we should note that it uses liquid electrolytes. The design methodologies and real demonstrations with energy-dense liquid-based systems are well-known, but those of solid-state batteries are almost rare. The microstructure of the two systems is completely different, which also affects the macro architecture and the practical fabrication process, so we want to emphasize that our demonstration is still meaningful in delivering the universal design logic for various SSBs.

At least for the initial performance, our pouch cell data can be a significant benchmark compared to the previous SSB literature, which showed the highest energy density (281 Wh kg^{-1}) and capacity (1 Ah) at the cell level, including whole packages in the total weight of the pouch-type SSBs under practical electrode design (94 wt% of AM in the electrode composition which is the highest AM ratio reported to date for SSB electrodes and high loading weight of 26 mg cm^{-2}) with the largest stacking number of SSB electrodes ever reported. Afterward, we could obtain a pouch cell data with higher energy density (310 Wh kg^{-1}) with capacity (0.5 Ah) at the cell level under practical electrode design (94 wt% of AM in the electrode composition and high loading weight of 30 mg cm^{-2}). Accordingly, we added this data in **Fig. 6c-d** and moved the original data of pouch cell with energy density (203 Wh kg^{-1}) from **Fig. 6a-b** to **Supplementary Fig. 11**. In addition, the data of figure demonstrating the performance benchmarking of SSBs against recently published studies was also changed.

Moreover, this paper focused on the cell engineering perspective by systematic design approach for high-loading-density electrodes and multi-stacked cell manufacturing necessary to achieve high energy density practical SSBs. This paper proposed the design logic covering multiscale and multi-parameters from the particle microstructures to the practical cell-based macro architecture.

Original:

Fig. 5: Electrochemical characteristics and compositions of practical SSB pouch cells

First charge–discharge electrochemical profiles and weight compositions of all the configuring materials of SSB pouch cells with (a–b) 156 mAh scale, 200 Wh kg⁻¹, and (c–d) 1 Ah scale, 280 Wh kg⁻¹. The 156 mAh- and 1 Ah-scale cell are composed of two and ten bilayer electrodes, respectively, and three and eleven freestanding thin Li-metal anodes, respectively. The testing was conducted within the operating voltage range of 2.7–4.3 V Li/Li⁺, and the condition was set to 0.05C under 45 °C. The composition included the entire passive packages.

Revised:

Fig. 6: Electrochemical characteristics and compositions of practical SSB pouch cells

Initial charge–discharge electrochemical profiles and weight compositions of all the configuring materials of SSB pouch cells with **(a–b) 1 Ah scale, 280 Wh kg^{-1} , and (c–d) 0.5 Ah scale, 310 Wh kg^{-1}** . The 1 Ah- and 0.5 Ah-scale cell are composed of four and ten bilayer electrodes, respectively, and three and eleven freestanding thin Li-metal anodes, respectively. The testing was conducted within the operating voltage range of 2.7–4.3 V Li/Li^+ , and the condition was set to 0.05C under 45 °C. The composition included the entire passive packages.

Original: Fig. 5a presents the first electrochemical charge/discharge profile for the pouch cell with an energy density over 200 Wh kg^{-1} . ~ The gravimetric composition of the pouch cell is depicted in Fig. 5b. ~ Compared to Fig. 5b, the fraction of the AM was boosted to almost 43 wt%, whereas that of the other inactive parts was decreased. This is because of the scaled-up electrode size and stacks of the SSB pouch cell, and the importance of prototyping toward practicality is highly significant.

Revised: **Supplementary Fig. 11a** presents the **initial** electrochemical charge/discharge profile for the pouch cell with an energy density over 200 Wh kg^{-1} . ~ The gravimetric composition of the pouch cell is depicted in **Supplementary Fig. 11b**. ~ Compared to **Supplementary Fig. 11b**, the fraction of the AM was boosted to almost 43 wt%, whereas that of the other inactive parts was decreased. This is because of the scaled-up electrode size and stacks of the SSB pouch cell, and the importance of prototyping toward practicality is highly significant. **Furthermore, the highest energy density of 310 Wh kg^{-1} of SSB pouch cell could be obtained with the higher total loading weight of 30 mg cm^{-2} , equivalent to an areal discharge capacity of 5.24 mAh cm^{-2} , as shown in Fig. 6c-d.**

Original:

Fig. 6: Performance benchmarking of SSBs against recently published studies

Revised:

Fig. 7: Performance benchmarking of SSBs against recently published studies

Continued from the comment of Reviewer #1:

1-2. There are some minor concerns with methodologies, which are generally academic standards (e.g., the use of stainless steel for the anodic stability ; use of very low current for Li/Li plating tests; these are hardly representative of the real conditions observed during cycling). It would be useful to observe, for example, the critical current density for this SPE.

→ **Response:** We appreciate your valuable comment. In regard to the use of stainless steel for anodic stability, Al current collector is enough to cover the operating voltage range under 4.3 V versus Li. Therefore, we used the Al as a current collector for the anodic side.

The applied current density (0.05 C, equivalent to 0.19 mA cm^{-2}) for pouch-type SSBs in this paper was low as you mentioned. The reason for the low current density in this paper is attributed to the thick cathodes and the large number of stacked electrodes inducing high overpotential in the pouch cells. To deconvolute the overpotentials arising from both the anode and cathode sides, we intentionally aligned the current densities between the pouch cells and Li/Li symmetric cells so that we could verify the certain polarization from the Li metal anode.

We also tested the additional rate capability with increasing by 0.05 C per every five cycles from 0.1C (0.3 mA cm^{-2}) to 0.25 C (0.75 mA cm^{-2}) for constant areal capacity (0.15 mAh cm^{-2}). The experimental method and result data were added in the revised manuscript and **Supplementary Fig. 5**. Based on the result, the critical current density was 0.25 C (0.75 mA cm^{-2}) with cell shortening. This result can stem from several reasons such as 1) chemically unstable SPE (due to dehydrofluorination during cycling) or 2) inhomogeneous charge-transfer between Li metal and SPE.

Further studies to reduce the overall cell resistance and improve the cyclability of SSBs should be done to solve the issues of limitations of current densities in terms of not only materials research but also cell fabrications in future works. We hope this explanation can relieve the reviewer's concern.

Supplementary Fig. 5: The rate performance of Li symmetric cycling test of Li/SPE/Li coin cell with constant areal capacity (0.15 mAh cm^{-2}) and different current densities ($\pm 0.3 \text{ mA cm}^{-2}$ for 30 mins (for 1-5 cycles), $\pm 0.45 \text{ mA cm}^{-2}$ for 20 mins (for 6-10 cycles), $\pm 0.6 \text{ mA cm}^{-2}$ for 15 mins (for 11-15 cycles), and 0.75 mA cm^{-2} for 12 mins (for 16-19 cycles)).

Discussion of Supplementary Fig. 5: The rate capability test of symmetric cells of Li/SPE/Li coin cell was performed with different current densities. Based on the result, there was a short-circuit at 0.25 C (0.75 mA cm^{-2}). This result can stem from several reasons such as 1) chemically not optimized SPE (due to dehydrofluorination of SPE during cycling) or 2) insufficient interfacial stability between lithium metal and SPE. To solve this problem, further optimization of SPE components in terms of both chemistry and engineering should be studied in future work.

Continued from the comment of Reviewer #1:

1-3. Other very minor concerns are related to a generally unclear description of the fabrication of electrodes, e.g., if those are simply cut, how these highly defective layers would behave in case of welding procedures, or, more generally, what kind of quality control is used to make the small 10 layer pouch (rejects, etc., for each part).

→ **Response:** Thank you for your comment. The more detailed fabrication of the cathode composite using WIP method was added in the manuscript and **Supplementary Fig. 20** in SI. A more detailed

explanation of the fabrication process of pouch-type cell was added in the manuscript and **Supplementary Fig. 21** in SI.

Page 29, line 1 (Supplementary Information)

Supplementary Fig. 20: Fabrication process of membrane-electrode assembly (MEA) using WIP method

a, Laminated cathode and SPE membrane. **b**, covered with parchment paper to avoid fluttering during vacuum sealing. Vacuum-packed before **(c)** and after WIP. **(d and e)**. **f**, membrane-electrode assembly (cathode-SPE-Li metal anode).

Discussion of Supplementary Fig. 20: As the first step, the cathode and SPE were compressed by the WIP separately. The WIP method can make not only uniform interface formation for the cathode but also squeeze and enhance the mechanical strength of SPE. After that, the compressed cathode and SPE were laminated and compressed again using the WIP method to maximize the interfacial contact between the cathode and SPE. The WIP method consists of three steps. The first step is increasing the pressure to 5500 bar gradually for 5 mins. Second, the pressure was maintained at 5500 bar for 5 mins. Finally, the pressure was reduced to zero gradually for 10 mins. After the WIP, the Li metal anode can be simply put onto the SPE of laminated cathode-SPE composite.

Supplementary Fig. 21: The scheme of cathode-SPE-anode with different sizes.

Discussion of Supplementary Fig. 21: The anode is larger than the cathode because the anode should take lithium ions from the cathode during the charging. SPE is larger than the electrodes to prevent the edge shortening.

Original: Li/SPE/NCM622 coin-cell (CR2032) tests were performed using different areal electrode loadings. One test was performed with a low areal electrode loading of 3.50 mg cm^{-2} , and another test was performed with a high areal electrode loading of 22.67 mg cm^{-2} .

Revised: Li/SPE/NCM622 coin-cell (CR2032) tests were performed using different areal electrode loadings. One test was performed with a low areal electrode loading of 3.50 mg cm^{-2} , and another test was performed with a high areal electrode loading of 22.67 mg cm^{-2} . The prepared SPE was used as

the solid electrolyte. The cathodes and SPE were prepared using the WIP. The first step is applying the WIP for each cathode and SPE separately. After that, the second step is applying the WIP for membrane-electrode assembly (MEA) which is laminated the cathode and SPE for maximizing MEA interfacial contact. The detailed explanation and figures can be seen in **Supplementary Fig. 20**. Li foil was used as the anode for all tests.

Page 27, line 18

Original: The same SPE and Li metal were used as the solid electrolyte and anode, respectively.

Revised: The same SPE and Li metal were used as the solid electrolyte and anode, respectively. **Here,** the sizes for the anode, cathode, and SPE were different: Anode is larger than the cathode because the anode should take lithium ions from the cathode during the charging. SPE is larger than the electrodes to prevent edge shortening (the schematic figure can be seen in **Supplementary Fig. 21**). The electrodes were cut by electrode punching machines with the designed sizes for anode and cathode. SPE was cut by a cutter. The MEA (membrane-electrode assembly: laminated cathode-SPE) was made by WIP which is the same fabrication process in the coin cell test above for interfacial contact. After that, the electrode tabs and the lead tab terminals were welded. Al pouch film was molded into the electrode shape and the depth was adjusted by the stopper with different heights depending on the number of stacks. To seal Al pouch film, the tab and side were sealed with dedicated machines. A vacuum sealing was introduced to the pouch cell as a final assembly, followed by hot-pressing to enhance the layer contact inside the pouch cell by applying constant heat and pressure.

Continued from the comment of Reviewer #1:

1-4. The main concern is that the paper is limited, focusing only on geometrical parameters. This can significantly underestimate the material required to fabricate a real cell or the chemical nature of said materials.

→ **Response:** We appreciate the reviewer's critical comment. It is a fact that extrinsic parameters such as geometrics were mainly considered for the rational design of the high-energy-density SSBs in this manuscript. It is because the morphology of the AM and SE materials are normally almost quasi-spherical even if they have different chemistries, so this common exterior facilitates building geometric-based versatile design principles which possible to simply guide the electrode composition, density, and loading. We regard this aspect of structural design as being underestimated for the multiscale design of practical SSBs. However, we also focused on not only extrinsic engineering parameters but also intrinsic characteristics for proposing the design guidelines for high-energy-density SSBs. For example, we considered the true densities of AM and SE materials to calculate the ionic areal specific resistance (ASR) in the composite particle and electrode density. In addition, the physical characteristics of the SE membrane such as true density, and ionic conductivity, were also considered for calculating the energy density at the cell level of SSBs. We believe this design approach is effective and novel, therefore, the claim of our research is noteworthy in numerical methodology to build the SSB electrode and cell design.

Continued from the comment of Reviwer #1:

1-5. A proper analysis should consider electrochemical variables and other types of resistances to guide material selection and cell design. For example, other types of resistance (e.g., diffusion coefficients, charge transfer) and measured variables (electrode tortuosity, coulombic efficiency, more in-depth description of N/P ratio).

→ **Response:** Thank you for your constructive comment. In response to the reviewer's feedback, we have measured the resistive properties of the composite electrodes such as diffusion coefficient and charge-transfer behavior based on GITT and EIS data as represented in **Supplementary Fig. 13** and **15**. The diffusion coefficient of the electrode ranged from 10^{-11} to 10^{-9} $\text{cm}^2 \text{V}^{-1} \text{s}^{-1}$. The charge-transfer resistance showed about 300 to 500 Ωcm^2 . However, we should note that those characteristics are also

combinational results dependent on intrinsic material properties and extrinsic electrode design because those results are highly variable to the design parameters even if we use the same materials. For example, the charge-transfer resistance was variable to the electrode loading level although the materials and compositions were completely identical. The diffusion coefficient was also changed as a function of SOC, so it was difficult to extract specific values for design guidelines. The electrode tortuosity, coulombic efficiency, and N/P ratio are also other extrinsic factors and highly dependent on the electrode and cell design parameters. Thus, we are unsure to conclude that these results would be other types of guidelines for materials selection in SSB cell design. We have to clarify that this manuscript is trying to deliver rational and general design methodologies for SSB materials integration into the electrodes and practical cells, not to evaluate whether particular materials are better for that or not. However, we highly agree that developing microscopic factors of materials as novel input parameters might open pathways toward unprecedented multiscale design logic, which are far beyond the scope of this research but must be done soon.

Supplementary Fig. 13: Coin-cell performance of SSBs based on SPE

a, Charge–discharge curves of Li/SPE/NCM622 coin cell with low cathode loading (3.50 mg cm^{-2}).

b, Charge–discharge curves of Li/SPE/NCM622 coin cell with high cathode loading (22.67 mg cm^{-2}).

c, Discharging capacity during cycling of Li/SPE/NCM622 coin cells with different cathode loading

levels. **d**, EIS results of Li/SPE/NCM622 coin cells with different cathode loading levels.

Page 25, line 1 (Supplementary Information)

Supplementary Fig. 16: Galvanostatic intermittent titration technique (GITT) result and the calculated diffusivity. a, Charge-discharge curve of GITT for Li/SPE/NCM622 coin-cell and b, calculated diffusivity from GITT at each relaxed point.

Page 21, line 5

Original: Unfortunately, the other electrochemical properties such as the cyclabilities are not relatively superior at this stage for various possible reasons. In **Supplementary Fig. 8**, these characteristics were slightly improved by detuning the electrode loading parameters, confirmed in coin half-cell format.

Revised: Unfortunately, the other electrochemical properties such as the cyclabilities are not relatively superior at this stage for various possible reasons. In **Supplementary Fig. 13**, these characteristics were slightly improved by detuning the electrode loading parameters, confirmed in coin half-cell format. **Here, NCM622 was chosen as a AM material instead of NCM811 even though the specific capacity is lower because of better chemical stability (Supplementary Fig. 14 and 15). Furthermore, the galvanostatic intermittent titration technique (GITT) result of coin-cell was shown in Supplementary Fig. 16 to confirm the comparable diffusivity ($\sim 10^{-9} \text{ cm}^2 \text{ S}^{-1}$), the intrinsic property, in our system compared with the general LIB system. We got acceptable diffusivity values⁵⁰⁻⁵², and it means there is no problem with AM caused by the discrepancy of diffusivity with the LIB system.**

Continued from the comment of Reviewer #1:

1-6. It is also unclear why the authors selected a SN-based membrane, given the poor cathodic stability - it is unclear what happens after decomposition.

→ **Response:** Thank you for your comment. Succinonitrile (SN) is a common plasticizer widely used for SPEs, and we are focusing on the universal SSB cell design principles toward high-energy-density, so we intentionally used general formulation of SPE to highlight our design logic. According to the related reference paper⁴, the authors developed a new type of elastomeric electrolyte consisting of PEGDA, butyl acrylate (BA), succinonitrile (SN), and LiTFSI. Here, SN is a representative plasticizer that enhances ionic conductivity due to the lowering polymer crystallinity as well as complexed with Li salts. The LSV result for the used SPE showed high oxidative stability (4.75 V), allowing for stable operation with NMC622 cathode within the electrochemical windows below 4.3 V vs. Li/Li⁺. It indicates that SN with Li salt is stable at high voltage. However, SN is commonly known to be unstable at lower voltage, so we observed the N 1s XPS data of Li metal anode after full cell cycling of Li/SPE/NCM622 coin cell to see the consequences of SN decomposition. The result showed the formation of Li₃N on the Li metal anode as plotted in **Supplementary Fig. 17** in SI and explanation in the manuscript.

Original:

Supplementary Fig. 9: XPS data of Li-metal anode after full-cell cycling of Li/SPE/NCM622 coin cell.

Discussion of Supplementary Fig. 9: After the full-cell cycling of the coin cell, XPS data of the Li-metal anode side was collected, and formation of C-F and LiF, stemming from the PVDF-HFP in the SPE, was observed. The LiF formed the stable layer on the Li-metal anode^{S2}.

Revised:

Supplementary Fig. 17: F 1s XPS and N 1s XPS data of Li-metal anode after full-cell cycling of Li/SPE/NCM622 coin cell.

Discussion of Supplementary Fig. 17: After the full-cell cycling of the coin cell, F1s XPS data of the

Li-metal anode side was collected, and formation of C-F and LiF, stemming from the PVDF-HFP in the SPE, was observed. In addition, N1s XPS data of the Li-metal anode after cycling could show the formation of Li₃N, stemming from the SN decomposition, was observed. The LiF and Li₃N formed the stable layer on the Li metal anode^{S2}.

Page 21, line 14

Original: XPS spectra of the Li-metal anode and FTIR spectra of SPE after the cycling test were measured, and the results indicated that stable LiF was formed with almost no structural change in SPE during cycling (**Supplementary Figs. 8 and 9**).

Revised: XPS spectra of the Li-metal anode and FTIR spectra of SPE after the cycling test were measured, and the results indicated that stable LiF and Li₃N were formed with almost no structural change in SPE during cycling (**Supplementary Figs. 17 and 18**).

Continued from the comment of Reviewer #1:

1-7. Considering there is more than one commercial software available for cell design and simulation, this approach is not competitive. The authors should consider expanding the scope for publishing in a journal like Nature Communications.

→ **Response:** Your comment encourages us to keep improving, and we are thankful for that. We want to emphasize that we are trying to provide simple design rules for practical SSBs easily adaptable by general researchers and engineers without high-performing and expensive computational solutions. Similarly, the current basic parameters used for simple electrode design, such as composition, loading level, and density, might have been first invented by anonymous individuals at a primitive stage of battery research. The battery cell could be designed based on those average parameters without simulation software, as it has been until now. We also hope that our design principles proposal helps SSB development toward practical applications.

In this manner, we devise the ideal binary model system to represent the average structure of AM and SE. For SSB electrode design, AM and SE are the main phases relatively compared to the small amount of conducting agent (CA) and binder. Therefore, we intentionally exclude the regard of CA and binder to simplify the model from the quaternary system. We admit that this proposed model has limitations in terms of electronic percolations as well as real particle size distribution. However, this model represents the simplified average structure, so this might be a cornerstone for building the complex structure of the advanced design toward practical SSBs.

To verify this logic, we compared our designed model (CCP based structure, monodispersed PSD) and realistic digital-twin model (polydispersed PSD) as well as experimental demonstrations for each 4 sections of the quadrant in **Fig. 1h** divided by balance and percolation thresholds. The comparative data results show that the two models are relatively well-matched to the experimental data in terms of voltage profiles, which indicates that those models are representable for the real SSB electrode microstructure. It means that our simple design logic and model are quite effective and comparable to the complex digital-twin model as shown in **Supplementary Fig. 6a**. To concrete our manuscript, we added **Fig. 2**, **Supplementary Fig. 6-7**, and **Supplementary Table 4-5**, and explanations in the manuscript accordingly.

Page 9, line 23

High fidelity of design principles verified by simulations and experiment

To validate our electrode design principles, we systemetically compared the electrochemical characteristics between experimental and simulated results based on our simplified ideal model and the realistic complex digital-twin model reflecting particle size distribution (PSD). Validation was representatively proceeded within the condition for each 4 sections of the quadrant in Fig. 1h divided by balance and percolation thresholds as shown in Fig. 2: (a) AM-poor and well-percolated, (b) AM-rich and well-percolated, (c) AM-poor and poor-percolated, and (d) AM-rich and poor-percolated. We

used a commercialized BatteryDict program to verify the design principles. As a result, all cases show quite analogous electrochemical voltage profile and charge capacity between experimental and simulation results as shown in **Fig. 2** and **Supplementary Fig. 6a**. We should note slight discrepancies in discharge capacity because BatteryDict was not available to simulate the initial irreversible capacity accompanying such as SEI formation. Please remind that the proposed ideal binary model system representing only AM and SE based on the core-shell building block and CCP arrangement does not reflect the electronic percolation network and real particle size distribution with the quaternary system including conducting agent and binder. However, on average, this model reliably represents the real electrode microstructure excluding the relatively negligible conducting agent and binder domain (content of carbon < 4.5 vol%). This indicates that those models are representable for the real SSB electrode microstructure on average, and our electrode model is quite effective and comparable to the complex digital-twin model and the real electrode microstructure.

Over the balance threshold, AM-rich cases (**Fig. 2b, d**) demonstrated higher energy density than AM-poor cases (**Fig. 2a, c**) as shown in **Supplementary Fig. 7**. This arises from the higher amount of cathode AM integrated into the same CCP-structure in AM-rich cases, thus resulting in higher energy density. Meanwhile, AM-poor cases excelled in power density by exhibiting the higher capacities (177.15 and 190.26 mAh g⁻¹) than AM-rich cases (165.79 and 175.83 mAh g⁻¹) under the same current density (36 mA g⁻¹). For the percolation threshold, the porosity should be lower than 26 vol% to provide the ionic percolation network in the core-shell CCP model as shown in **Fig. 1e**. The well-percolated cases (**Fig. 2a, b**) show the lower porosity translating to higher electrode density and greater capacity. This result indicates that the more efficient ionic transport pathway is provided for well-percolated systems due to the densification. Although the SE network covering AM cannot be percolated in the core-shell model case in the condition where the porosity is over 26 vol%, interestingly, the electrochemical operation is feasible for real experiment and polydispersed case even with higher electrode porosity. It is because the polydispersed AM and SE contacted each other, and

SE did not evenly cover AM like the core-shell model in real case. Given that all the selected model systems represent carbon and binder-deficient electrode, poorly percolated cases (Fig. 2c, d) show lower capacity than well-percolated cases resulting from poorly connected ionic transport interfaces.

Despite the limitation of ideal assumption that can be applicable to the case over the percolation threshold, this proposed principle is quite compelling in delivering quantitative design guidelines that have not been presented before. This cross-validation, involving both modeling and experimentation, underscores the reliability of our design principle for high-energy-density and high-power-density oriented SSBs. In addition, this simplified principle can rationally guide the direction of electrode design without complex calculations. Based on this, it could build a universal and multi-dimensional design space by a combination of various design parameters.

Fig. 2: Design principle validation through experimentation and simulations

Cross-validation of electrochemical characteristics from monodispersed and polydispersed PSD digital-twin models and experimental complex electrode microstructure. **a**, AM-poor and well-percolated, **b**, AM-rich and well-percolated, **c**, AM-poor and poor-percolated, **d**, AM-rich and poor-percolated cases, and **e**, converted electrode density landscape of Fig.1h to represent of balance and percolation thresholds in orthogonal coordinates. Color of voltage profiles indicate the apparent electrode density for each case.

Modeling of 3D composite cathode structures

NCM622, SPE, and Super P were used in varying weight ratios (NCM622: SPE: Super P = 94.0: 4.0: 2.0 and 78.5: 19.5: 2.0 wt%) to create a virtual composite cathode. Particle size distribution of spherical NCM622 through particle size analysis (PSA, Malvern Panalytical, Mastersizer 3000) was applied for modeling and the monodispersed CCP structure was generated using a single particle size (8.55 μm). NCM622 was covered by SPE as a core-shell structure, following the suggested design principle. Each virtual 3D structure had a voxel length of 0.2 μm and was modeled under design conditions with domain sizes, thickness and electrode densities as shown in **Supplementary Table 4**. All these processes were conducted using the GrainGeo and ProcessGeo modules of the GeoDict 2023 software (Math2Market, Germany).

Charge/discharge simulation of 3D composite cathode structures

An electrochemical simulation was performed using the Left Identity Right (LIR) solver after composing a digital-twin solid-state cell through the Design battery in the BatteryDict module. The pouch cell is operated in the same condition as the pouch cell test. Each cell in four regions was charged at 0.1C and discharged at 0.2C ($1\text{C} = 180 \text{ mA g}^{-1}$) at 45 °C in the cut off voltage range of 2.7–4.3 V shown in **Fig 2**. The input parameters for this simulation are detailed in **Supplementary Table 5**.

Supplementary Fig. 6: **a**, Comparison electrochemical characteristics with the models and experiments (experimental, monodispersed and polydispersed PSD models) of SSB pouch cell in Fig. **6a**, **b**, the particle size distribution of NCM622 used in modeling, and the visualized electrode structure for **c**, polydispersed and **d**, monodispersed PSD models.

Supplementary Fig. 7: Energy density comparison divided by composite for each case. Case a-b are well-percolated cases and Case c-d are poor-percolated cases in each AM-poor and AM-rich conditions, respectively.

Supplementary Table 4. 3D digital twin structure information.

AM:SE (exclude CA)		Value	Porosity (%)	Thickness (μm)	Domain size ($\mu\text{m}\times\mu\text{m}\times\mu\text{m}$)	Specific capacity (mAh g^{-1})
96:4 (Pouch cell)	Well- percolation	Experiment	14.92	70.9	-	173.16
		Polydispersed PSD	15.16	70.8	48.4 \times 48.4 \times 70.8	-
		Monodispersed PSD	15.23	72.0	48.4 \times 48.4 \times 72.0	-
96:4 (AM rich)	Poor- percolation	Experiment	35.38	51.6	-	165.79
		Polydispersed PSD	35.55	51.6	48 \times 48 \times 51.6	-
		Experiment	14.00	42.0	-	175.83
96:4 (AM rich)	Well- percolation	Polydispersed PSD	14.26	42.0	48 \times 48 \times 42	-
		Monodispersed PSD	15.23	42.0	48 \times 48 \times 42	-
		80:20 (AM poor)	Poor- percolation	Experiment	26.94	49.2
Polydispersed PSD	27.05			49.2	50.0 \times 50.0 \times 49.2	-
Experiment	15.52			34.2	-	190.26
80:20 (AM poor)	Well- percolation	Polydispersed PSD	15.27	34.2	55.2 \times 55.2 \times 34.2	-
		Monodispersed PSD	14.74	34.6	55.2 \times 55.2 \times 34.6	-

Supplementary Table 5. Modeling parameters.

Parameters	Value
Intrinsic electronic conductivity of NCM622 at 45°C (S cm ⁻¹)	9.79 × 10 ⁻⁴
Intrinsic lithium diffusion coefficient of NCM622 at 45°C (m ² s ⁻¹)	1.53 × 10 ⁻¹³
Maximum lithium concentration of NCM622 (mol m ⁻³)	4.92 × 10 ⁴
Intrinsic ionic conductivity of SPE at 45°C (S cm ⁻¹)	1.25 × 10 ⁻³
Intrinsic lithium diffusion coefficient of SPE at 45°C (m ² s ⁻¹)	4.20 × 10 ⁻¹¹
Lithium transference number of SPE	0.90
Lithium concentration of SPE (mol m ⁻³)	845

Continued from the comment of Reviewer #1:

1-8. The results show a relatively poor cycle life, including for the 1 Ah pouch (first cycle being ca. 280 Wh/kg, second one from the third-party being about 265 Wh/kg), an aspect that is not sufficiently considered. To prove their point (that their method is effective) extended cycling over at least a hundred of cycles should be demonstrated.

→ **Response:** We are grateful for your pointing out. We should clarify that our research does not focus on the specific chemistries optimization of SSBs. Still, it focuses on providing the universal design principles to achieve high energy density for practical SSBs by systematic and multiscale approaches and fabrication methodologies, and those characteristics at the pouch cell level are for demonstrating our design guidelines. We were also convinced that the initial performance of our SSB pouch cell was enough to prove the effectiveness of our design logic, which was the first demonstration of Ah-scale and multi-stacked SSB pouch cell for scientific peer review. We do believe that the cycle characteristics of pouch cells are not relative to the proposed design strategies. This is because performance factors such as energy density and power density can be calculated, but the cycle characteristics are not predictable without recent big data-based machine learning tools.

Nevertheless, we admit that cyclability should be better for practical usage even if it is not in the scope of our research. The cycle performance of our cell slightly improves when operated at a higher C-rate than 0.05C (We added the data in **Supplementary Fig. 19**). The cell designed by our design logic can achieve several cycle numbers although it is a coin cell level. However, the capacity suddenly drops after about 150~160 hours because the dehydrofluorination of PVDF-HFP increases over time. So, we tried to suppress the dehydrofluorination with several methods such as washing, annealing the cathode material, and lowering the operation temperature to counteract the effects of residual lithium compounds on the cathode surface and elevated temperature. The electrochemical performances from all tries were still suddenly dropped after a similar time passed. Therefore, we agree that the optimizing chemical compatibility is necessary to achieve a more extended cycle life and a longer operation time. However, given that the cell shows an improved cycle number (10 cycles) by controlling the charge/discharge protocol, our design logic is a proper guide for designing the electrode. Even though the cyclability is insufficient in this study, it is noteworthy that the energy density of pouch-type SSBs could be achieved well, similar to the calculated energy density based on the design logic considering the most ideal case. This study can provide insight into the design of the various parameters from electrodes to the cell fabrications and operating conditions to achieve the desired energy density of SSBs with various types of AM and SE with different chemistries.

Page 21, line 17

Original: However, this performance is still not comparable to that of LIBs, potentially because of a possible chemical side reaction, the well-known dehydrofluorination for PVDF-based polymers⁵⁰.

Revised: However, this performance is still not comparable to that of LIBs, potentially because of a possible chemical side reaction, the well-known dehydrofluorination for PVDF-based polymers⁵⁰

(Supplementary Fig. 3), and increased resistance because of high cathode loading weight and a large number of electrodes in the pouch-type stack. To suppress the dehydrofluorination, several methods

such as washing, annealing the cathode material, and lowering the operation temperature to counteract the effects of residual lithium on the cathode surface and elevated temperature, were tried. The electrochemical performances from all tries are still suddenly dropped after approximately 150~160 hours. However, given that the cell shows an improved cycle number (10 cycles) by controlling the charge/discharge protocol, our design logic is a proper guideline for designing the electrode, as shown in Supplementary Fig. 19.

Page 28, line 1 (Supplementary Information)

Supplementary Fig. 19: Cycle performance of Li/SPE/NCM622 coin cells under different C-rate protocols. 0.05C operation at 45°C with drying (a) and annealing (b) DI water of washed cathode, c, 0.1C charge / 0.2C discharge at RT with drying cathode, and d, electrochemical voltage vs. time profiles during cycling of three cases before the sudden capacity fading.

Continued from the comment of Reviewer #1:

1-9. Finally, given that their article would be open-access, the authors should consider sharing a spreadsheet or file to facilitate the improvement of their cell design model.

→ **Response:** Thank you for your comment. We also agree that sharing the design toolkit of this research will help the readers' understanding, and it might be further enhanced if we open the source code in this article with respect to the open-access intent of *Nature Communications*. Reflecting on the reviewer's suggestion, we provide our revised cell design spreadsheet, so-called "SolidXCell" which is interactive to easily visualize the comprehensive structure with intertwined multiple parameters that aid in understanding the design landscape. Our "SolidXCell" is a dynamic toolkit for solid-state battery design, merging 'Solid' for robustness and focus with 'XCell,' a subtle nod to Excel, with the versatility denoted by 'X.' It signifies a comprehensive approach to battery design represented by the dual reference to battery 'cells' and Excel spreadsheet 'cells.' This spreadsheet is easy to understand and edit, so we hope this logic is widely adopted and further advanced by the following peers of the battery community. The key features of this cell design toolkit are below.

1. Inventory Sheet

- **Library of Materials:** This sheet serves as a comprehensive database of materials used in SSB cell design. It includes various categories like anodes, cathodes, electrolytes, binders, and conducting agents.
- **User Interaction:** Users have the flexibility to add new materials to this library. This feature allows for the expansion and customization of the database to suit specific project needs.
- **Dynamic Linking:** A crucial feature of this sheet is the ability to link the inventory data to other sheets related to cell design. This is done through a dedicated "INVENTORY" button located at the A1 cell. It is essential for users to use this feature to ensure that the data across the spreadsheet remains interconnected and updated.

INVENTORY	NAME	Charge mAh/g	Cutoff voltage V vs. Li/Li+	Discharge mAh/g	Cutoff voltage V vs. Li/Li+	Nominal voltage V vs. Li/Li+	Coulombic efficiency %	True density g/cm3	Ionic conductivity mS/cm	Areal weight g/m2	Weight mg	porosity vol%
Anode	Li	-	0.00	3860.00	0.00	0.00	#VALUE!	0.53	-	-	-	-
Anode	Graphite	372.00	0.00	360.00	3.00	0.10	96.77	2.26	-	-	-	-
Binder	PVDF	-	-	-	-	-	-	1.77	-	-	-	-
Binder	SBR	-	-	-	-	-	-	1.04	-	-	-	-
Binder	CMC	-	-	-	-	-	-	1.60	-	-	-	-
Binder	NBR	-	-	-	-	-	-	1.35	-	-	-	-
Binder	PTFE	-	-	-	-	-	-	2.20	-	-	-	-
Cathode	LCO	155.56	4.30	140.00	3.00	3.70	90.00	5.12	-	-	-	-
Cathode	NCM111	172.22	4.30	155.00	3.00	3.70	90.00	4.83	-	-	-	-
Cathode	NCM622	194.44	4.30	175.00	3.00	3.70	90.00	4.76	-	-	-	-
Cathode	NCM622 (KIER-SSB)	205.56	4.30	185.00	3.00	3.70	90.00	4.76	-	-	-	-
Cathode	NCM811	222.22	4.30	200.00	3.00	3.70	90.00	4.79	-	-	-	-
Cathode	LFP	161.11	4.30	145.00	3.00	3.40	90.00	3.68	-	-	-	-
Cathode	S	-	-	-	-	-	#DIV/0!	3.68	-	-	-	-
Conducting agent	Super P	-	-	-	-	-	-	1.95	-	-	-	-
Current collector	Al	-	-	-	-	-	-	2.70	-	-	-	-
Current collector	Cu	-	-	-	-	-	-	8.96	-	-	-	-
Electrolyte	LPSCl	-	-	-	-	-	-	1.54	1.00	-	-	-
Electrolyte	LLZO	-	-	-	-	-	-	5.01	1.00	-	-	-
Electrolyte	Li3InCl6	-	-	-	-	-	-	2.59	1.00	-	-	-
Electrolyte	1M LiPF6/EC/EMC/VC	-	-	-	-	-	-	1.20	1.00	-	-	0.00
Electrolyte	PVDF-HFP/LITFSI/SN	-	-	-	-	-	-	1.80	1.00	-	-	0.00
Lead tab	Ni tab	-	-	-	-	-	-	-	-	-	-	98.30
Lead tab	Al tab	-	-	-	-	-	-	-	-	-	-	36.90
Package	Al pouch film	-	-	-	-	-	-	-	-	180.00	-	-
Separator	PVDF-HFP/LITFSI/SN	-	-	-	-	-	-	1.80	1.00	-	-	0.00
Separator	PE	-	-	-	-	-	-	0.98	1.00	-	-	33.00

2. Internal Resistance Sheet

- Composite Electrode Design:** This sheet is dedicated to the particle-level design of composite electrodes using a core(AM)-shell(SE) model. This approach is fundamental in optimizing the internal resistance of the battery cells.
- Material Selection and Composition:** Users can select materials from a predefined drop-down menu and input gravimetric compositions. The sheet is programmed to automatically calculate the corresponding volumetric composition, leveraging each material's true density.
- Geometric Calculations:** Additional functionalities include the calculation of shell thickness based on the average radius of active material (AM) and the determination of areal-specific resistance by inputting the pressed shell thickness.

Parameter	Unit	AM	CA	BINDER	EL	Sum
		NCM622	Super P	PVDF	PVDF-HFP	
Material true density	g/cc	4.76	1.95	1.77	1.80	-
Gravimetric composition	wt%	88.27	0.00	0.00	11.73	100.00
Volumetric composition	vol%	74.00	0.00	0.00	26.00	100.00
Composite electrode true density	g/cc					3.99
Core-shell size	um	5.00			0.53	
Pressed shell thickness	um				0.53	
Swelled shell thickness	um				0.55	
Specific capacity	mAh/g	175.00				
Ionic conductivity	mS/cm				1.00	
Building block	Areal loading	mg/cm2	2.60	0.00	0.00	2.94
	Areal specific capacity	mAh/cm2	0.45			
	Areal specific resistance	ohm cm2				8.27

3. Cell Design Sheet

- **Cell-Level Design Framework:** This sheet facilitates the design of SSB cells at the macro level, considering various components and materials.
- **Extensive Material Selection:** A wide range of materials can be selected from multiple drop-down menus, allowing for a detailed and customized cell design.
- **Advanced Calculations:** The sheet is equipped to handle complex calculations like converting gravimetric to volumetric composition and determining electrode density. These calculations are guided by critical thresholds (e.g., balance threshold for AM:SE ratio, percolation threshold for porosity).
- **Visualization and Analysis Tools:** Users can create dynamic illustrations of the cell design in both top and side views. Additionally, the sheet automatically generates pie-chart diagrams for energy density compositions, providing a visual representation of the cell's energy distribution.

Electrode	Anode	Cathode	Electrolyte	TOP-VIEW	SIDE-VIEW				
Active material	Li 100.00 wt% 100.00 vol%	NCM622 10.00 μm 94.00 wt% 85.88 vol%	True density 0.00 g/cm ³ Excess ratio 0.00 vol% E/C ratio 0.00 g/Ah Volume 0.00 cm ³						
Electrolyte	0.00 wt% 0.00 vol%	PVDF-HFP/LITFSI/SN 4.00 wt% 9.66 vol%	Weight 0.00 g Separator PVDF-HFP/LITFSI/SN True density 1.80 g/cm ³ Pore volume 0.00 cm ³ Porosity 0.00 vol% Areal ionic bulk resistance 7.50 Ω cm ²						
Binder	0.00 wt% 0.00 vol%	0.00 wt% 0.00 vol%	Thickness 75.00 μm Edge margin 1.50 mm Width 3.30 cm Height 4.45 cm Area 14.69 cm ²						
Conducting agent	Super P 0.00 wt% 0.00 vol%	Super P 2.00 wt% 4.46 vol%	Volume 0.11 cm ³ Weight 0.20 g						
Initial charge capacity	- mAh/g - mAh/cm ²	209.35 mAh/g 6.05 mAh/cm ²	Cell specification Parallel stack 8 unit cell Nominal discharge voltage 3.76 V C-rate 0.05 C Nominal discharge capacity 491.99 mAh Ionic bulk resistance 2.74 Ω Width 3.90 cm Height 5.45 cm Thickness 1.76 mm Volume 2.73 cm ³ Weight 5.77 g						
Initial discharge capacity	3860.00 mAh/g 4.12 mAh/cm ²	181.21 mAh/g 5.24 mAh/cm ²	Cell composition			wt%	vol%		
Initial coulombic efficiency	- %	86.56 %	AM			2.715 g	47.0%	0.689 cm ³	27.8%
N/P	1.94		EL			0.116 g	2.0%	0.078 cm ³	3.1%
Discharge nominal voltage	0.00 V vs. Li/Li ⁺	3.76 V vs. Li/Li ⁺	B			0.000 g	0.0%	0.000 cm ³	0.0%
Areal ionic bulk resistance	Ω cm ²	250.38 Ω cm ²	CA			0.058 g	1.0%	0.036 cm ³	1.4%
Areal loading weight	1.07 mg/cm ²	30.74 mg/cm ²	CC			0.262 g	4.5%	0.097 cm ³	3.9%
Spatial loading density	0.53 g/cm ³	3.60 g/cm ³	AM			0.120 g	2.1%	0.224 cm ³	9.1%
True loading density	0.53 g/cm ³	4.35 g/cm ³	E	0.000 g	0.0%	0.000 cm ³	0.0%		
Pore volume	0.00 cm ³	0.02 cm ³	B	0.000 g	0.0%	0.000 cm ³	0.0%		
Porosity	0.00 vol%	17.22 vol%	CA	0.000 g	0.0%	0.000 cm ³	0.0%		
Thickness	20.00 μm	85.39 μm	Volume	0.000 g	0.0%	0.000 cm ³	0.0%		
Edge margin	0.50 mm	- mm	Weight	0.018 g	0.3%	0.002 cm ³	0.1%		
Width	3.00 cm	2.90 cm	Cell specification	EL	0.000 g	0.0%	0.000 cm ³	0.0%	
Height	4.15 cm	4.05 cm	Power density	SEP	1.586 g	27.5%	0.881 cm ³	35.6%	
Area	12.45 cm ²	11.75 cm ²	Energy density	PF	0.765 g	13.3%	0.468 cm ³	18.9%	
Volume	0.02 cm ³	0.10 cm ³		LT	0.135 g	2.3%	-	-	
Weight	0.01 g	0.36 g							
Current collector	Cu	Al							
True density	8.96 g/cm ³	2.70 g/cm ³							
Thickness	0.00 μm	20.00 μm							
Width	3.00 cm	2.90 cm							
Height	4.15 cm	4.05 cm							
Area (+ tab)	0.40 cm ²	12.15 cm ²							
Volume	0.00 cm ³	0.02 cm ³							
Weight	0.00 g	0.07 g							
Lead tab	Ni tab	Al tab							
Width	4.00 mm	4.00 mm							
Height	10.00 mm	10.00 mm							
Weight	98.30 mg	36.90 mg							

Reviewer #2

This article presented rational design principles for achieving high energy density in practical solid-state lithium batteries. The author demonstrated the feasibility of this principle through the fabrication of a 10-layer stacked solid-state pouch cell with a solid polymer electrolyte, resulting in an energy density of 280 Wh kg^{-1} and 600 Wh L^{-1} . Authors need to address the following comments before this paper can be considered further.

→ **Response:** We are very grateful for the reviewer's encouraging comment. As the reviewer advised, we carefully addressed the comments with point-by-point responses to reinforce the manuscript as follows and hope the manuscript is now ready for publication in timely manner.

Continued from the comment of Reviewer #2:

2-1. Page 6 line 15-19: the ionic ASR inside of AM particles (during $0\sim 5\mu\text{m}$ of core-compressed surface distance) showed different trends between fig 1 b and fig 1 c, please give explanations.

→ **Response:** Thank you for your comment. The trend of the ionic ASR inside the core(AM)-shell(SE) cathode composite particles between **Fig. 1b** and **Fig. 1c** is different because the thickness of SE shells is different according to the designated AM/SE ratio. **Fig. 1b** represents the AM-poor and SE-rich cases typical among most previous SSB literature. Because of the thick shell with rich SE, the composite-particulate ASR will be lower with its broader cross-section area. However, **Fig. 1c** shows balanced AM/SE case with volumetric ratio of 74:26 with CCP structure, which means it has thinner SE shell with relatively higher ASR compared to **Fig. 1b**. We should also notice that if the composite particulate electrodes are compressed with pole-to-pole direction if SE is highly ductile, then the SE shell might be deformed and slightly get thicken by their constant volume so that ASR will be decreased by the spanned interfacial area with plane contact. This explanation was added in the manuscript.

Original: The AM-poor, SE-rich condition represents most recently published research articles (Fig. 1b); however, they must be tailored to have a balanced AM/SE ratio as shown in Fig. 1c to achieve higher energy density.

Revised: The AM-poor, SE-rich condition represents most recently published research articles (Fig. 1b); however, they must be tailored to have a balanced AM/SE ratio as shown in Fig. 1c to achieve higher energy density. **The trend of the ionic ASR inside the cathode composite particles between Fig. 1b and Fig. 1c is inversely proportional to the SE composition because it governs the thickness of the SE shell.**

Continued from the comment of Reviwer #2:

2-2. Page 6 line 6-8: “the threshold is based on an ideal assumption that does not consider the actual particle morphology and size distribution”, in fact, the size distribution and mechanical property of active cathode particles can affect the dense packing state of cathode plates and thus the battery energy-density. The author needs to take these factors into account in the model and modify it to make it more close to the actual situation.

→ **Response:** We appreciate your feedback. We agree with the reviewer’s point about the importance of physical characteristics such as particle size distribution and mechanical properties of AM in affecting the packing states. However, in this study, we focused on proposing the simplified design tool under the most ideal case. This study is sufficiently meaningful to guide how to design the various key parameters for achieving the desired energy density of SSBs because there is no study yet about the general systematic design logics for high energy-density of SSBs in the current state. We want to emphasize that we are trying to provide simple design rules for practical SSBs easily adaptable by general researchers and engineers without high-performing and expensive computational solutions.

Similarly, the current basic parameters used for simple electrode design, such as composition, loading level, and density, might have been first invented by anonymous individuals at a primitive stage of battery research. The battery cell could be designed based on those average parameters without simulation software, as it has been until now. We also hope that our design principles proposal helps SSB development toward practical applications.

In this manner, we devise the ideal binary model system to represent the average structure of AM and SE. For SSB electrode design, AM and SE are the main phases relatively compared to the small amount of conducting agent (CA) and binder. Therefore, we intentionally exclude the regard of CA and binder to simplify the model from the quaternary system. We admit that this proposed model has limitations in terms of electronic percolations as well as real particle size distribution. However, this model represents the simplified average structure, so this might be a cornerstone for building the complex structure of the advanced design toward practical SSBs.

To verify this logic, we compared our designed model (CCP based structure, monodispersed PSD) and realistic digital-twin model (polydispersed PSD) as well as experimental demonstrations for each 4 sections of the quadrant in **Fig. 1h** divided by balance and percolation thresholds. The comparative data results show that the two models are relatively well-matched to the experimental data in terms of voltage profiles, which indicates that those models are representable for the real SSB electrode microstructure. It means that our simple design logic and model are quite effective and comparable to the complex digital-twin model as shown in **Supplementary Fig. 6a**. To concrete our manuscript, we added **Fig. 2**, **Supplementary Fig. 6-7**, and **Supplementary Table 4-5**, and explanations in the manuscript accordingly.

Page 9, line 23

High fidelity of design principles verified by simulations and experiment

To validate our electrode design principles, we systemetically compared the electrochemical

characteristics between experimental and simulated results based on our simplified ideal model and the realistic complex digital-twin model reflecting particle size distribution (PSD). Validation was representatively proceeded within the condition for each 4 sections of the quadrant in **Fig. 1h** divided by balance and percolation thresholds as shown in **Fig. 2**: (a) AM-poor and well-percolated, (b) AM-rich and well-percolated, (c) AM-poor and poor-percolated, and (d) AM-rich and poor-percolated. We used a commercialized BatteryDict program to verify the design principles. As a result, all cases show quite analogous electrochemical voltage profile and charge capacity between experimental and simulation results as shown in **Fig. 2** and **Supplementary Fig. 6a**. We should note slight discrepancies in discharge capacity because BatteryDict was not available to simulate the initial irreversible capacity accompanying such as SEI formation. Please remind that the proposed ideal binary model system representing only AM and SE based on the core-shell building block and CCP arrangement does not reflect the electronic percolation network and real particle size distribution with the quaternary system including conducting agent and binder. However, on average, this model reliably represents the real electrode microstructure excluding the relatively negligible conducting agent and binder domain (content of carbon < 4.5 vol%). This indicates that those models are representable for the real SSB electrode microstructure on average, and our electrode model is quite effective and comparable to the complex digital-twin model and the real electrode microstructure.

Over the balance threshold, AM-rich cases (**Fig. 2b, d**) demonstrated higher energy density than AM-poor cases (**Fig. 2a, c**) as shown in **Supplementary Fig. 7**. This arises from the higher amount of cathode AM integrated into the same CCP-structure in AM-rich cases, thus resulting in higher energy density. Meanwhile, AM-poor cases excelled in power density by exhibiting the higher capacities (177.15 and 190.26 mAh g⁻¹) than AM-rich cases (165.79 and 175.83 mAh g⁻¹) under the same current density (36 mA g⁻¹). For the percolation threshold, the porosity should be lower than 26 vol% to provide the ionic percolation network in the core-shell CCP model as shown in **Fig. 1e**. The well-percolated cases (**Fig. 2a, b**) show the lower porosity translating to higher electrode density and greater

capacity. This result indicates that the more efficient ionic transport pathway is provided for well-percolated systems due to the densification. Although the SE network covering AM cannot be percolated in the core-shell model case in the condition where the porosity is over 26 vol%, interestingly, the electrochemical operation is feasible for real experiment and polydispersed case even with higher electrode porosity. It is because the polydispersed AM and SE contacted each other, and SE did not evenly cover AM like the core-shell model in real case. Given that all the selected model systems represent carbon and binder-deficient electrode, poorly percolated cases (Fig. 2c, d) show lower capacity than well-percolated cases resulting from poorly connected ionic transport interfaces.

Despite the limitation of ideal assumption that can be applicable to the case over the percolation threshold, this proposed principle is quite compelling in delivering quantitative design guidelines that have not been presented before. This cross-validation, involving both modeling and experimentation, underscores the reliability of our design principle for high-energy-density and high-power-density oriented SSBs. In addition, this simplified principle can rationally guide the direction of electrode design without complex calculations. Based on this, it could build a universal and multi-dimensional design space by a combination of various design parameters.

Fig. 2: Design principle validation through experimentation and simulations

Cross-validation of electrochemical characteristics from monodispersed and polydispersed PSD digital-twin models and experimental complex electrode microstructure. **a**, AM-poor and well-percolated, **b**, AM-rich and well-percolated, **c**, AM-poor and poor-percolated, **d**, AM-rich and poor-percolated cases, and **e**, converted electrode density landscape of Fig.1h to represent of balance and percolation thresholds in orthogonal coordinates. Color of voltage profiles indicate the apparent electrode density for each case.

Modeling of 3D composite cathode structures

NCM622, SPE, and Super P were used in varying weight ratios (NCM622: SPE: Super P = 94.0: 4.0: 2.0 and 78.5: 19.5: 2.0 wt%) to create a virtual composite cathode. Particle size distribution of spherical NCM622 through particle size analysis (PSA, Malvern Panalytical, Mastersizer 3000) was applied for modeling and the monodispersed CCP structure was generated using a single particle size (8.55 μm). NCM622 was covered by SPE as a core-shell structure, following the suggested design principle. Each virtual 3D structure had a voxel length of 0.2 μm and was modeled under design conditions with domain sizes, thickness and electrode densities as shown in **Supplementary Table 4**. All these processes were conducted using the GrainGeo and ProcessGeo modules of the GeoDict 2023 software (Math2Market, Germany).

Charge/discharge simulation of 3D composite cathode structures

An electrochemical simulation was performed using the Left Identity Right (LIR) solver after composing a digital-twin solid-state cell through the Design battery in the BatteryDict module. The pouch cell is operated in the same condition as the pouch cell test. Each cell in four regions was charged at 0.1C and discharged at 0.2C ($1\text{C} = 180 \text{ mA g}^{-1}$) at 45 °C in the cut off voltage range of 2.7–4.3 V shown in **Fig 2**. The input parameters for this simulation are detailed in **Supplementary Table 5**.

Supplementary Fig. 6: **a**, Comparison electrochemical characteristics with the models and experiments (experimental, monodispersed and polydispersed PSD models) of SSB pouch cell in **Fig. 6a**, **b**, the particle size distribution of NCM622 used in modeling, and the visualized electrode structure for **c**, polydispersed and **d**, monodispersed PSD models.

Supplementary Fig. 7: Energy density comparison divided by composite for each case. Case a-b are well-percolated cases and Case c-d are poor-percolated cases in each AM-poor and AM-rich conditions, respectively.

Supplementary Table 4. 3D digital twin structure information.

AM:SE (exclude CA)		Value	Porosity (%)	Thickness (μm)	Domain size ($\mu\text{m}\times\mu\text{m}\times\mu\text{m}$)	Specific capacity (mAh g^{-1})
96:4 (Pouch cell)	Well- percolation	Experiment	14.92	70.9	-	173.16
		Polydispersed PSD	15.16	70.8	48.4×48.4×70.8	-
		Monodispersed PSD	15.23	72.0	48.4×48.4×72.0	-
96:4 (AM rich)	Poor- percolation	Experiment	35.38	51.6	-	165.79
		Polydispersed PSD	35.55	51.6	48×48×51.6	-
		Experiment	14.00	42.0	-	175.83
96:4 (AM rich)	Well- percolation	Polydispersed PSD	14.26	42.0	48×48×42	-
		Monodispersed PSD	15.23	42.0	48×48×42	-
		80:20 (AM poor)	Poor- percolation	Experiment	26.94	49.2
Polydispersed PSD	27.05			49.2	50.0×50.0×49.2	-
Experiment	15.52			34.2	-	190.26
80:20 (AM poor)	Well- percolation	Polydispersed PSD	15.27	34.2	55.2×55.2×34.2	-
		Monodispersed PSD	14.74	34.6	55.2×55.2×34.6	-

Supplementary Table 5. Modeling parameters.

Parameters	Value
Intrinsic electronic conductivity of NCM622 at 45°C (S cm ⁻¹)	9.79 × 10 ⁻⁴
Intrinsic lithium diffusion coefficient of NCM622 at 45°C (m ² s ⁻¹)	1.53 × 10 ⁻¹³
Maximum lithium concentration of NCM622 (mol m ⁻³)	4.92 × 10 ⁴
Intrinsic ionic conductivity of SPE at 45°C (S cm ⁻¹)	1.25 × 10 ⁻³
Intrinsic lithium diffusion coefficient of SPE at 45°C (m ² s ⁻¹)	4.20 × 10 ⁻¹¹
Lithium transference number of SPE	0.90
Lithium concentration of SPE (mol m ⁻³)	845

Continued from the comment of Reviewer #2:

2-3. Why does the electrolyte thickness reach nearly 100 μm ? Can the thickness of SE be reduced to further improve the energy density? In previous works, the thickness of electrolytes have been reduced to below 20 μm . The thick SE selected in this paper is not representative of high- energy-density batteries.

[1] Toluene Tolerated $\text{Li}_{9.88}\text{GeP}_{1.96}\text{Sb}_{0.04}\text{S}_{11.88}\text{Cl}_{10.12}$ Solid Electrolyte toward Ultrathin Membranes for All-Solid-State Lithium Batteries. *Nano Lett.* 2023, **23**, 1, 227–234.

[2] An Ultra-Thin Crosslinked Carbonate Ester Electrolyte for 24 V Bipolar Lithium-Metal Batteries. *J. Electrochem. Soc.* 2022, **169**, 090509.

[3] Scalable, Ultrathin, and High-Temperature-Resistant Solid Polymer Electrolytes for Energy-Dense Lithium Metal Batteries. *Adv. Energy Mater.* 2022, **12**, 2103720.

→ **Response:** We are thankful for your feedback. We actually tested the SPE membrane with 40 μm thickness as an SE separator for Li/SPE/NCM622 coin cell cycling to enhance the cell-level energy density. However, the cell-shortening occurred during the initial charge, and we observed the uneven contact between the SE membrane and the cathode with lithium dendrite formation after the cell disassembly. So we attached MEA without lithium metal anode via WIP until 550 MPa, and it was confirmed that the charge voltage profile was relatively stabilized, but still had the cell shortening with tiny lithium dendrites. Therefore, we speculated that the uniformly high pressure was not the solution and the thin SE membrane was insufficient to accommodate the high areal capacity of around 4 mAh cm^{-2} of lithium electroplating due to the inhomogeneous charge-transfer. We further thickened the SE membrane up to 80 μm and finally had a stable electrochemical cycling profile without cell shortening as shown in **Supplementary Fig. 10**. We agree that the thickness of the SE membrane is quite crucial to the energy density of SSBs as plotted in **Fig. 5i**, but we should highlight that the SE should be carefully designed depending on the amount of charge per unit area of the thick cathode. The suggested references by the reviewer claimed that they utilized thin SE membranes, but those had very low AM-

loaded cathode (only 2~3 mg cm⁻²)^{5,6} or not even mention the areal capacity and AM loading⁶.

Page 18, line 1 (Supplementary Information)

Supplementary Fig. 10: Coin-cell performance of SSBs using different thicknesses of SE membranes

a, Charge–discharge curves of Li/SPE/NCM622 coin cell with thin SE membrane (40 μm) without WIP, thin (40 μm) and thick (80 μm) SE membranes compressed with WIP. **b**, Photo of disassembled coin cells with 40 μm thick SE membrane without (b) and with WIP (c).

Discussion of Supplementary Fig. 10: The SPE membrane with 40 μm thickness was tested as an SE separator for Li/SPE/NCM622 coin cell cycling to enhance the cell-level energy density. However, the cell-shortening occurred during the initial charge, and it was observed the uneven contact between the SE membrane and the cathode with lithium dendrite formation after the cell disassembly. So MEA without lithium metal anode was attacked via WIP until 550 MPa, and it was confirmed that the charge voltage profile was relatively stabilized, but still had the cell shortening with tiny lithium dendrites. Therefore, it was speculated that the uniformly high pressure was not the solution and the thin SE membrane was insufficient to accommodate the high areal capacity of around 4 mAh cm⁻² of lithium electroplating due to the inhomogeneous charge-transfer. The further thickened SE membrane up to 80 μm had a stable electrochemical cycling profile without cell shortening. The thickness of the SE membrane is quite crucial to the energy density of SSBs as plotted in Fig. 5i, but those should be

carefully designed depending on the amount of charge per unit area of the thick cathode.

Page 17, line 19

Original: The thickness is tunable; however, we intentionally fabricated a slightly thick membrane (98 μm) to accommodate the amount of charge from the electrode.

Revised: The thickness is tunable; however, we intentionally fabricated a slightly thick membrane (75~100 μm) to accommodate the amount of charge from the electrode. To see the effect of the thickness of SE membrane on the cycling stability, the coin cell performance using different thicknesses of SE membranes was shown in **Supplementary Fig. 10**. This data could prove that the thicker SE membrane is conditionally necessary for operating the cell with high loading cathodes.

Continued from the comment of Reviewer #2:

2-4. The purpose of this paper is to prepare high-energy-density pouch cells, why not choose materials with higher energy density (such as NCM811) as the cathode active material?

→ **Response:** We appreciate the reviewer's feedback. We had tested Li/SPE/NCM811 configured coin cells, but those were not well-operated for unspecified reasons. When we tried to laminate MEA composed of NCM811 as an AM, the SE membranes turned brownish, the yield rate of those cells was low, and some were not cycled well as shown in **Supplementary Fig. 14**. The issues of SE membrane color change were also found in the degraded cell with NCM622 after cycle test in **Supplementary Fig. 19**. We assumed that those were related to dehydrofluorination of PVDF based polymers which is well-known to this chemical under the high pH conditions based on the several supporting data. According to this, we also specified that the residual lithium compounds, such as Li_2CO_3 or LiOH on the surface of NCM811, triggered the degradation of SPE and were found to have low electrochemical cycling behavior. We agree that if the chemical stability of PVDF polymer based-SPE is increased, then we can further enhance the SSB energy density by utilizing Ni-rich NCM cathodes. We considered that those efforts are beyond the scope of this research, so we concluded that they should be separately studied for the future works. We added the explanation in the manuscript and **Supplementary Fig. 14 and 15** in SI including the additional references [S12, S13]^{8,9}.

Supplementary Fig. 14: Charge–discharge curves of Li/SPE/NCM811 coin cell performance.

Discussion of Supplementary Fig. 14: Li/SPE/NCM811 configured coin cells were not well-operated for unspecified reasons. The laminate MEA composed of NCM811 as an AM, the SE membranes turned brownish, the yield rate of those cells was low, and some were not discharged well as shown in this figure. The issues of SE membrane color change were also found in the degraded cell with NCM622 after cycle test in **Supplementary Fig. 19**. We assumed that those were related to dehydrofluorination of PVDF based polymers which is well-known to this chemical under the high pH conditions^{S12} based on the several supporting data. According to this, we also specified that the residual lithium compounds, such as Li₂CO₃ or LiOH on the surface of NCM811, triggered the degradation of SPE and were found to have low electrochemical cycling behavior.

Original:

S11. Lee, M. J., Han, J., Lee, K., Lee, Y. J., Kim, B. G., Jung, K. N., Kim, B. J. & Lee, S. W.

Elastomeric electrolytes for high-energy solid-state lithium batteries. *Nature* **601**, 217-222 (2022).

Revised:

S11. Lee, M. J., Han, J., Lee, K., Lee, Y. J., Kim, B. G., Jung, K. N., Kim, B. J. & Lee, S. W.

Elastomeric electrolytes for high-energy solid-state lithium batteries. *Nature* **601**, 217-222 (2022).

S12. Seong, W. M., Cho, K. H., Park, J. W., Park, H., Eum, D., Lee, M. H., Kim, I. S., Lim, J. & Kang, K. Controlling residual lithium in high-nickel (> 90%) lithium layered oxides for cathodes in lithium-ion batteries. *Angew. Chem. Int. Ed.* **59**, 18662-18669 (2020).

→ added to Ref. S12

S13. Taguet, A., Ameduri, B. & Boutevin, B. Crosslinking of vinylidene fluoride-containing fluoropolymers. In: *Crosslinking in Materials Science. Adv. Polym. Sci.* **184**, 127-211 (2005).

→ added to Ref. S13

Page 24, line 1 (Supplementary Information)

Supplementary Fig. 15: SEM images of the cathodes (NCM622 or NCM811) before and after Li/SPE/NCM622 or Li/SPE/NCM811 coin cell tests.

SEM images of NCM622 cathode **a**, before, and **b**, after Li/SPE/NCM622 coin cell cycling test. SEM images of NCM811 cathode **c**, before, and **d**, after Li/SPE/NCM811 coin cell cycling test. The scale bars are 20 μm .

Page 21, line 5

Original: Unfortunately, the other electrochemical properties such as the cyclabilities are not relatively superior at this stage for various possible reasons. In **Supplementary Fig. 8**, these characteristics were slightly improved by detuning the electrode loading parameters, confirmed in coin half-cell format.

Revised: Unfortunately, the other electrochemical properties such as the cyclabilities are not relatively superior at this stage for various possible reasons. In **Supplementary Fig. 13**, these characteristics were slightly improved by detuning the electrode loading parameters, confirmed in coin half-cell format. **Here, NCM622 was chosen as a AM material instead of NCM811 even though the specific capacity is lower because of better chemical stability (Supplementary Fig. 14 and 15).**

Reviewer #3

The manuscript is focused on the engineering, and the WIP technique is not applicable in the cathode manufacturing process. Additionally, some key information is missing, also there are too many mistakes in manuscript. Thus, I am not in favor of publishing this article in *Nature Communications*. The quality is not close to the standard of this journal.

→ **Response:** We deeply appreciate the reviewer's critical comment. We admit that our manuscript focuses on the engineering aspects of SSB prototyping compared to the other previous literature. Thus, this manuscript might not provide novel insight in terms of SSB materials research. However, we believe that balanced research efforts should be conducted not only on materials but also on cell design as well as process development because the materials are no longer the only limiting factor for SSB prototyping. The journal to which we submitted this manuscript is interdisciplinary *Nature Communications*, and we also know that the editor has already discerned the importance of this practical aspect as seen in this review paper¹¹. Therefore, we are confident that our research can shed light on the unnoticed area and provide other novel inspirations to the community.

We should clarify that our key points are on the rational quantitative design rule toward practical SSBs relying on electrode and cell structures, not just on the fabrication processing techniques. We believe that our perspective is highly related to the viewpoints from several recent papers such as Prof. Janek group¹, Prof. Meng group¹⁰, etc. Reminding that the Battery 500 Consortium led by PNNL cast some valuable engineering insight toward high-energy-density lithium metal batteries, we also hope to commit to the community by delivering our design logic for developing practical high-energy-density SSBs.

Continued from the comment of Reviewer #3:

3-1. The transference number of the polymer electrolyte is 0.9, which means that the transportation of anions is selectively restricted. And this value is higher than that of most reported works. The authors had better provide a more detailed explanation, maybe NMR experiment is need. Additionally, the method to EIS fitting and the calculation method to the transference number must be provided.

→ **Response:** Thank you for the reviewer's feedback. We evaluated the lithium transference number of SPE with ion-blocking cell configuration by the Bruce-Vincent method which is a standard protocol to identify the portion of cation and anion movement within the electrolyte, and we do not doubt the technique of our measurement. Still, as a consequence, our specially formulated SPE shows a remarkable lithium transference number value of about 0.9 compared to most of the previous peer-reviewed literature as shown in **Supplementary Fig. 2b**. We should also inform the reviewer that this SPE shows very low activation energy of about 100 meV as well in **Supplementary Fig. 2a**. We have also noticed that this indicates the almost single-ion conducting behavior of our SPE, to be honest, but it is not possible to identify the specific reason for these characteristics at this stage even if we just used common ingredients for SPEs such as PVDF-HFP, LiTFSI, and SN.

Based on our SPE formulating recipe and the TGA data in **Supplementary Fig. 2f**, the coordination ratio between Li^+ cation and SN molecule was about 1:8. It seems to exceed the reported coordination number of 2~3 depending on the temperature¹², which might ensure we had almost full ion dissociation to form Li^+/SN solvation clusters. We speculate this facilitated the solid-state cation conduction with high ion conductivity and low activation energy within the matrix coupling to the uncoordinated rotational SN plastic crystal domain. On the other hand, the TFSI⁻ anion group might have relatively sluggish conducting kinetics due to the larger solvation cluster within the SPE, especially when interacting with non-crystalline polar polymers such as PVDF. Intertwined with lowering crystallinity by incorporating SN plasticizer and HFP copolymer, the SPE might have better segmental motion

essential to the ion conduction, so we assume those results were observed.

We should mention that the origin of the SPE characteristics must be investigated by the following studies, but in this manuscript, we request that attention be paid to our novel quantitative design rule for practical SSB electrode and cell design as well as their demonstration rather than its chemistries. To relieve the reviewer's concern, we provide additional explanations for the method of EIS fitting with all data (as seen in **Supplementary Fig. 2a, 2b, and 9d**). The calculation of the activation energy and transference number was also explained in the revised manuscript.

Page 25, line 19

Original: In addition, EIS spectra of the SS/SPE/SS cell were collected at various temperatures (30°C, 40°C, and 80°C) to calculate the activation energy (E_a) of the SPE.

Revised: In addition, EIS spectra of the SS/SPE/SS cell were collected at various temperatures (30°C, 40°C, and 80°C) to calculate the activation energy (E_a) of the SPE. **The E_a can be calculated by the Arrhenius equation (Equation (1)).**

Equation (1):

$$\sigma T = \sigma_0 \exp\left(-\frac{E_a}{RT}\right)$$

(Here, σ is the ionic conductivity of the sample, T is the temperature of conductivity measurement, R is the gas constant, and E_a is the activation energy. The slope of the Arrhenius plots changed to a lesser extent depending upon the composition of the solid electrolyte and temperature; hence the activation energy is directly proportional to the slope.)

Page 25, line 22

Original: The lithium-ion transference number was derived using the chronoamperometry (CA) method and EIS spectra measured before and after CA measurement of the Li/SPE/Li cell.

Revised: The lithium-ion transference number was derived using the **Bruce-Vincent method based on chronoamperometry (CA) and EIS spectra measured before and after CA measurement of the Li/SPE/Li cell.** **The lithium-ion transference number can be calculated by Equation (2).**

Equation (2):

$$t_{\text{Li}^+} = \frac{I_{\text{ss}}(\Delta V - I_0 R_0)}{I_0(\Delta V - I_{\text{ss}} R_{\text{ss}})}$$

(Here, t_{Li^+} is the lithium transference number, I_0 and I_{ss} are the initial and steady-state currents, ΔV is the potential applied across the cell (here, this value was 100 mV.), and R_0 and R_{ss} are the initial and steady-state charge-transfer resistances SE membrane.)

Continued from the comment of Reviwer #3:

3-2. Some key experimental information and methods are missing. For example, test and calculation methods of activation energy, preparation of freestanding thin Li-metal anode, and so on.

→ **Response:** Thank you for your comment. A more detailed explanation of the fabrication of the cathode composite using WIP was added in the manuscript and **Supplementary Fig. 20**. The other explanation about the fabrication of pouch-type cells was added in the manuscript and **Supplementary Fig. 21**. Furthermore, the calculation of the activation energy and transference number was also explained in the revised manuscript.

Supplementary Fig. 20: Fabrication process of membrane-electrode assembly (MEA) using WIP method

a, Laminated cathode and SPE membranes. **B**, covered with parchment paper to avoid fluttering during vacuum sealing. Vacuum-packed before (**c**) and after WIP. (**d** and **e**). **f**, MEA (WIP laminated cathode-SPE, and lithium metal anode).

Discussion of Supplementary Fig. 20: As the first step, the cathode and SPE were compressed by the WIP separately. The WIP method can make not only uniform interface formation for the cathode but also squeeze and enhance the mechanical strength of SPE. After that, the compressed cathode and SPE were laminated and compressed again using the WIP to maximize the interfacial contact between the cathode and SPE. The WIP consists of three steps. The first step is to increase the pressure to 550 MPa gradually for 5 minutes. Second, the pressure was maintained for 5 minutes. Finally, the pressure was gradually reduced to zero for 10 minutes. After the WIP, the Li metal anode can be put onto the SPE of the laminated cathode-SPE composite.

Page 30, line 1 (Supplementary Information)

Supplementary Fig. 21: The scheme of cathode-SPE-anode with different sizes.

Discussion of Supplementary Fig. 21: The anode is larger than the cathode because the anode should take lithium ions from the cathode during the charging. SPE is larger than the electrodes to prevent the edge shortening.

Page 26, line 21

Original: Li/SPE/NCM622 coin-cell (CR2032) tests were performed using different areal electrode loadings. One test was performed with a low areal electrode loading of 3.50 mg cm^{-2} , and another test was performed with a high areal electrode loading of 22.67 mg cm^{-2} .

Revised: Li/SPE/NCM622 coin-cell (CR2032) tests were performed using different areal electrode loadings. One test was performed with a low areal electrode loading of 3.50 mg cm^{-2} , and another test was performed with a high areal electrode loading of 22.67 mg cm^{-2} . The prepared SPE was used as the SE membrane. The cathodes and SPE were prepared using the WIP. In the first step, WIP was

introduced separately for each cathode and SPE to have denser structure. After that, the same technique was repeated for membrane-electrode assembly (MEA) fabrication which laminated the cathode and SPE to maximize MEA interfacial contact. The detailed explanation and figures can be seen in **Supplementary Fig. 20**. Li foil was used as the anode for all tests.

Page 27, line 18

Original: The same SPE and Li metal were used as the solid electrolyte and anode, respectively.

Revised: The same SPE and Li metal were used as the solid electrolyte and anode, respectively. **Here,** the sizes for the anode, cathode, and SPE were different: Anode is larger than the cathode because the anode should take lithium ions from the cathode during the charging as well as regarding geometric N/P ratio. SPE was larger than the Li metal anodes to prevent edge shortening (the schematic figure can be seen in **Supplementary Fig. 21**). The electrodes were cut by punching machines with the designated sizes for anode and cathode. SPE was cut by a cutter for the target dimension. The MEA (membrane-electrode assembly: laminated cathode-SPE) was made by WIP which was the same fabrication process in the coin cell test above for interfacial contact. After that, the electrode tabs and the lead tab terminals were welded by ultrasonic method. Al pouch film was molded into the electrode shape and the depth was adjusted by the stopper with different heights depending on the number of stacks. To the seal Al pouch film, the tab and side were sealed with dedicated machines. A vacuum sealing was introduced to the pouch cell as a final assembly, followed by hot-pressing to enhance the layer contact inside the pouch cell by applying constant heat and pressure.

Page 25, line 19

Original: In addition, EIS spectra of the SS/SPE/SS cell were collected at various temperatures (30°C, 40°C, and 80°C) to calculate the activation energy (E_a) of the SPE.

Revised: In addition, EIS spectra of the SS/SPE/SS cell were collected at various temperatures (30°C,

40°C, and 80°C) to calculate the activation energy (E_a) of the SPE. **The E_a can be calculated by the Arrhenius equation (Equation (1)).**

Equation (1):

$$\sigma T = \sigma_0 \exp\left(-\frac{E_a}{RT}\right)$$

(Here, σ is the ionic conductivity of the sample, T is the temperature of conductivity measurement, R is the gas constant, and E_a is the activation energy. The slope of the Arrhenius plots changed to a lesser extent depending upon the composition of the solid electrolyte and temperature; hence the activation energy is directly proportional to the slope.)

Page 25, line 22

Original: The lithium-ion transference number was derived using the chronoamperometry (CA) method and EIS spectra measured before and after CA measurement of the Li/SPE/Li cell.

Revised: The lithium-ion transference number was derived using the **Bruce-Vincent method based on** chronoamperometry (CA) and EIS spectra measured before and after CA measurement of the Li/SPE/Li cell. **The lithium-ion transference number can be calculated by Equation (2).**

Equation (2):

$$t_{\text{Li}^+} = \frac{I_{\text{ss}}(\Delta V - I_0 R_0)}{I_0(\Delta V - I_{\text{ss}} R_{\text{ss}})}$$

(Here, t_{Li^+} is the lithium transference number, I_0 and I_{ss} are the initial and steady-state currents, ΔV is the potential applied across the cell (here, this value was 100 mV.), and R_0 and R_{ss} are the initial and steady-state charge-transfer resistances SE membrane.)

Continued from the comment of Reviewer #3:

3-3. The current density used in the plating-stripping tests of the Li||Li cell was too low (0.15 mA cm⁻²). The current should reach at least 2 mA cm⁻² for practical applications.

→ **Response:** We appreciate your valuable comment. The applied current density (0.05 C, equivalent to 0.19 mA cm⁻²) for pouch-type SSBs in this paper was low as you mentioned. The reason for the low current density in this paper is attributed to the thick cathodes and the large electrode stack number inducing high overpotential in the pouch cells. To deconvolute the overpotentials arising from both the anode and cathode sides, we intentionally aligned the current densities between the pouch cells and Li/Li symmetric cells so that we could verify the certain polarization from the Li metal anode.

We also tested the additional rate capability with increasing by 0.05 C per every five cycles from 0.1C (0.3 mA cm⁻²) to 0.25 C (0.75 mA cm⁻²) for constant areal capacity (0.15 mAh cm⁻²). The experimental method and result data were added in the revised manuscript and **Supplementary Fig. 5**. Based on the result, the critical current density was 0.25 C (0.75 mA cm⁻²) with cell shortening. This result can stem from several reasons such as 1) chemically unstable SPE (due to dehydrofluorination during cycling) or 2) inhomogeneous charge-transfer between Li metal and SPE.

Further studies to reduce the overall cell resistance and improve the cyclability of SSBs should be done to solve the issues of limitations of current densities in terms of not only materials research but also cell fabrications in future works. We hope this explanation can relieve the reviewer's concern.

Supplementary Fig. 5: The rate performance of Li symmetric cycling test of Li/SPE/Li coin cell with constant areal capacity (0.15 mAh cm^{-2}) and different current densities ($\pm 0.3 \text{ mA cm}^{-2}$ for 30 mins (for 1-5 cycles), $\pm 0.45 \text{ mA cm}^{-2}$ for 20 mins (for 6-10 cycles), $\pm 0.6 \text{ mA cm}^{-2}$ for 15 mins (for 11-15 cycles), and 0.75 mA cm^{-2} for 12 mins (for 16-19 cycles)).

Discussion of Supplementary Fig. 5: The rate capability test of symmetric cells of Li/SPE/Li coin cell was performed with different current densities. Based on the result, there was a short-circuit at 0.25 C (0.75 mA cm^{-2}). This result can stem from several reasons such as 1) chemically not optimized SPE (due to dehydrofluorination of SPE during cycling) or 2) insufficient interfacial stability between lithium metal and SPE. To solve this problem, further optimization of SPE components in terms of both chemistry and engineering should be studied in future work.

Page 7, line 7

Original: Moreover, FTIR spectroscopy analysis of the SPE and Li symmetric cycling tests using the SPE were also conducted, with the results presented in **Supplementary Figs. 2 and 3**.

Revised: Moreover, FTIR spectroscopy analysis of the SPE and Li symmetric cycling tests using the SPE were also conducted, with the results presented in **Supplementary Figs. 3-5**.

Continued from the comment of Reviewer #3:

3-4. Page 8, line 1: resistance would thus be minimized using this structure. The author had better give the resistance values with or without using this structure.

→ **Response:** Thank you for your comment. In this manuscript, we presumed that the most ideal occupancy of SE in the composite electrode should be in the interstitial site of AM, so their balanced volumetric ratio might be 74 vol% (AM) and 26 vol% (SE) according to the densest packed structure, so-called a cubic close-packed (CCP) arrangement (**Fig. 1a**). Besides, we also assume that the most ideal model of AM/SE composite electrode at the particle-level is core(AM)-shell(SE) structure because they have a maximized interfacial contact between those phases theoretically (**Fig. 1b-c**). Combining these hypothesises, the ideally integrated composite electrode would have AM with CCP structure filled with SE in the void space if the electrode porosity was 0 vol% (**Fig. 1g**). We claimed that the electrode resistance might be lowest with this structure because the interfacial contact area among the core-shell would be maximized as well as the broadening of a cross-sectional area without any voids. However, if the electrode porosity was 0~26 vol% (**Fig. 1e-f**), the interfacial contact area between the core-shell building block might be reduced and formed voids which would behave like resistance in terms of electrode performance. If the electrode porosity is over 26 vol% (**Fig. 1d**), then there are rich voids and the contact between the building blocks might be lost in our ideal model case, but those will be practically operated because the AM and SE are not evenly distributed and contacted each other. The series of **Fig.1d-g** implies that the importance of electrode density and porosity in terms of ionic percolation in SSB electrodes even if they have completely identical electrode composition. We should note that the exact value of resistance of certain electrode structures is not possible to specify due to the variables in the SE chemistries and AM combinations, but we delivered the comprehensive equations for calculating the ionic resistance in the particle and electrode level in **Supplementary Note 1** including **Equation 1-4**. We are confident that our ideal model assumption and the quantified rational logic for SSB electrode design are quite novel and can guide the following

studies correctly to achieve the desired characteristics toward practical SSBs. We hope our revised explanation can mitigate the reviewer’s concern and clarify the numerical key points of the logic.

Page 2, line 1 (Supplementary Information)

Original:

Supplementary Note 1. Calculation of ionic ASR of the particle composing the composite cathode

For calculation of the ionic ASR of the particle composing the composite cathode, there is the assumption that the particle has a core–shell structure. Here, the core and shell consist of the AM and SE, respectively.

Equation (1):

$$R = \rho \frac{l}{A}$$

(Here, R is the ionic ASR of the particle, ρ is the density of the particle, l is the length of the particle, and A is the area of the particle.)

Equation (2):

$$dR = \rho \frac{dl}{dA} = \rho \frac{r d\theta}{2\pi r \sin\theta dr} = \frac{\rho d\theta}{2\pi \sin\theta dr}$$

(Here, r is the radius of the particle, θ is the angle between the radius and the core–surface distance.)

Equation (3):

$$dR = \frac{\rho}{2\pi (r_2 - r_1)} \frac{d\theta}{\sin\theta} = \frac{\rho}{2\pi (r_2 - r_1)} \int_{\theta}^{\pi-\theta} \frac{d\theta}{\sin\theta}$$

(Here, r_2 is the radius of the core–shell particle, r_1 is the radius of the core side.)

Equation (4):

$$R = \frac{1}{2\pi\sigma (r_2 - r_1)} \ln\left(\frac{1 + \cos\theta}{1 - \cos\theta}\right) = \frac{1}{2\pi\sigma (r_2 - r_1)} \ln\left(\frac{r_2 + r_2\cos\theta}{r_2 - r_2\cos\theta}\right) = \frac{1}{2\pi\sigma (r_2 - r_1)} \ln\left(\frac{r_2 + x}{r_2 - x}\right)$$

(Here, x is the core–surface distance, which is $r_2\cos\theta$. $0 < \theta < \frac{\pi}{2}$, $0 < r_2\cos\theta < 1$.)

Revised:

Supplementary Note 1. Calculation of ionic ASR of the particle composing the composite cathode

For calculation of the ionic ASR of the particle composing the composite cathode, there is the assumption that the particle has a core-shell structure. Here, the core and shell consist of the AM and SE, respectively.

Equation (1):

$$R = \rho \frac{l}{A}$$

(Here, R is the ionic ASR of the **shell**, ρ is the **resistivity** of the **shell**, l is the **nominal** length of the **shell**, and A is the **cross-sectional** area of the **shell**.)

Supplementary Fig. 1: Schematic of ASR representation in SE shell with infinitesimal component

Equation (2):

$$dR = \rho \frac{dl}{dA} = \rho \frac{r d\theta}{2\pi r \sin\theta dr} = \frac{\rho d\theta}{2\pi \sin\theta dr}$$

(Here, r is the **spherical** radius of the **shell**, θ is the angle **from the pole to the compressed edge point at the spherical core**.)

Equation (3): integration of dR from θ to $\pi - \theta$ (pole-to-pole compression)

$$R = \frac{\rho}{2\pi (r_2 - r_1)} \int_{\theta}^{\pi-\theta} \frac{d\theta}{\sin\theta} = \frac{1}{2\pi\sigma (r_2 - r_1)} \int_{\theta}^{\pi-\theta} \frac{d\theta}{\sin\theta}$$

(Here, r_2 is the **outer** radius of the shell, r_1 is the **inner** radius of the **shell**, σ is the ionic conductivity of SE. $dr = r_2 - r_1$ is the thickness of shell, not infinitesimal.)

Equation (4):

$$R = \frac{1}{2\pi\sigma (r_2 - r_1)} \ln \left(\frac{1 + \cos\theta}{1 - \cos\theta} \right) = \frac{1}{2\pi\sigma (r_2 - r_1)} \ln \left(\frac{r_2 + r_2 \cos\theta}{r_2 - r_2 \cos\theta} \right) = \frac{1}{2\pi\sigma (r_2 - r_1)} \ln \left(\frac{r_2 + x}{r_2 - x} \right)$$

(Here, $x = r_2 \cos\theta$ and $r_2 - r_2 \cos\theta$ is the **compressed shell thickness**, where $0 < \theta < \frac{\pi}{2}$.)

We should note that the ASR of trace amounts to the near compressed edge (hatched area in **Supplementary Fig. 1**) was neglected due to the ease of calculations at the spherical coordinate system.

Continued from the comment of Reviwer #3:

3-5. Page 11, line 5: X-ray CT images are shown in Figure 2g-h, not Figure 1h.

→ **Response:** We are thankful for the pointing out. We apologize for the misspelling of the figure numbering. We revised it in the manuscript as per the reviewer's comment.

Page 13, line 5

Original: The measured porosity of the WIP-fabricated electrode is almost the same as the calculated value depending on the electrode density in the X-ray CT image in **Fig. 1h**; however, the electrode porosities for the other compressing methods differed dramatically.

Revised: The measured porosity of the WIP-fabricated electrode is almost the same as the calculated value depending on the electrode density in the X-ray CT image in **Fig. 3g-i**; however, the electrode porosities for the other compressing methods differed dramatically.

Continued from the comment of Reviwer #3:

3-6. Page 12, line 24: I don't understand the description of Fig. 1b, c. It Maybe Supplementary Fig. 7b, c?

→ **Response:** We are appreciate to the reviewer's comment. It means that the electrode loading weight or thickness can be calculated by converting target IR drop with leveraging the summed ASR of the unit building block of core-shell structure. By well-known Ohm's law, the ASR of the composite building blocks (shown in **Fig. 1b, c**, and **Supplementary Note 1**) can be used as a unit resistance term, and we can serieally connect this to quantify the overall IR of the electrode which can be converted to the electrode thickness and loading weight. In this regard, we mentioned **Fig. 1b, c** in this sentence, and the more detailed calculation process of how to calculate the electrode loading weight or thickness is shown in **Supplementary Note 3** using Ohm's law.

Page 13, line 1 (Supplementary information)

Original:

Supplementary Note 3. Calculation of the cathode IR drop of SSBs

Equation (6):

$$\text{IR drop} = Q_{AM,sp} \times \frac{\text{cathode loading thickness}}{2 \times r_{AM,sp}} \times ASR \times \frac{\text{cathode loading thickness}}{2 \times r_{AM,sp}} \times C$$

– rate

Here, IR drop is the voltage drop [mV]; $Q_{AM,sp}$ is the areal specific capacity of a single particle of AM [mAh cm⁻²], which can be calculated using **Equation (13)**; the *cathode loading thickness* [cm] can be calculated by dividing the areal cathode loading by the electrode density; $r_{AM,sp}$ is the radius of a single particle of AM [cm]; *ASR* is the areal specific resistance of the composite [Ohm cm²], which can be calculated using **Equation (1)**; and *C – rate* is the rate at which the battery is providing energy [C = 1/h].

Equation (7):

$$Q_{AM,sp} = \frac{Q_{AM} \times \text{true density}_{AM} \times V_{AM,sp}}{A_{composite}}$$

Here, $Q_{AM,sp}$ is the areal specific capacity of a single particle of AM [mAh cm^{-2}]; Q_{AM} is the specific capacity of the AM [mAh g^{-1}]; true density_{AM} is the material true density of the AM [g cm^{-3}]; $V_{AM,sp}$ is the volume of a single particle of AM [cm^3], which can be calculated by **Equation (14)**; and $A_{composite}$ is the area of the composite consisting of the AM and SE [cm^2], which can be calculated using **Equation (15)**.

Equation (8):

$$V_{AM,sp} = \frac{4}{3}\pi \times (r_{AM,sp})^3$$

Equation (9):

$$A_{composite} = \pi \times (r_{AM,sp} + t_{squeezed SE})^2$$

Here, $t_{squeezed SE}$ is the squeezed thickness of SE [cm].

Equation (10):

$$ASR = \ln \left(\frac{(r_{AM} + t_{squeezed SE}) + r_{AM}}{(r_{AM} + t_{squeezed SE}) - r_{AM}} \right) / (4 \times t_{squeezed SE} \times \sigma_{SE}) \times (r_{AM} + t_{squeezed SE})^2$$

Here, σ_{SE} is the ionic conductivity of the SE [S/cm].

Revised:

Supplementary Note 3. Calculation of the cathode IR drop of SSBs

Equation (6):

$$\begin{aligned}v_{\text{IR drop}} &= \left(Q_{\text{core-shell}} \times \text{C-rate} \times \frac{T_{\text{cathode}}}{2 \times r_{\text{AM}}} \right) \times \left(\text{ASR}_{\text{core-shell}} \times \frac{T_{\text{electrode}}}{2 \times r_{\text{AM}}} \right) \\ &= (J_{\text{core-shell}} \times n_{\text{core-shell}}) \times (\text{ASR}_{\text{core-shell}} \times n_{\text{core-shell}}) \\ &= J_{\text{total}} \times \text{ASR}_{\text{total}}\end{aligned}$$

Here, $v_{\text{IR drop}}$ is the IR drop [mV]; $Q_{\text{core-shell}}$ [mAh cm⁻²], $\text{ASR}_{\text{core-shell}}$ [Ohm cm²], $J_{\text{core-shell}}$, $n_{\text{core-shell}}$ are the areal specific capacity, the areal specific resistance, the current density, and the total number of units in the cathode of a core-shell unit building block composed of AM/SE, respectively; **C-rate** is the rate at which the battery is providing energy [C = 1/h]; T_{cathode} [cm] is the thickness of the cathode which can be calculated by dividing the areal cathode loading by the electrode density; r_{AM} [cm] is the radius of a single particle of AM; J_{total} [mA cm⁻²] and $\text{ASR}_{\text{total}}$ [Ohm cm²] are the current density and the areal specific resistance of the cathode, respectively.

Equation (7):

$$Q_{\text{core-shell}} = \frac{Q_{\text{AM}} \times d_{\text{AM}} \times V_{\text{AM}}}{A_{\text{core-shell}}}$$

Here, $Q_{\text{core-shell}}$ is the areal specific capacity of a core-shell unit building block composed of AM/SE [mAh cm⁻²]; Q_{AM} is the specific capacity of the AM [mAh g⁻¹]; d_{AM} is the material true density of the AM [g cm⁻³]; V_{AM} is the volume of a single particle of AM [cm³], which can be calculated by **Equation (14)**; and $A_{\text{core-shell}}$ is the area of the composite consisting of the AM and SE [cm²], which can be calculated using **Equation (15)**.

Equation (8):

$$V_{\text{AM}} = \frac{4}{3} \pi \times (r_{\text{AM}})^3$$

Equation (9):

$$A_{\text{core-shell}} = \pi \times (r_{\text{AM}} + t_{\text{swelled SE}})^2$$

Here, $t_{\text{swelled SE}}$ is the **swelled** thickness of SE after compression [cm].

Equation (10):

$$ASR = \frac{\ln \left(\frac{r_{\text{AM}} + t_{\text{swelled SE}}}{r_{\text{AM}}} \right) + r_{\text{AM}}}{(4 \times t_{\text{swelled SE}} \times \sigma_{\text{SE}}) \times (r_{\text{AM}} + t_{\text{swelled SE}})^2}$$

Here, σ_{SE} is the ionic conductivity of the SE [S/cm].

Continued from the comment of Reviwer #3:

3-7. The cycling performance of the cell is not enough.

→ **Response:** We are grateful for your pointing out. We should clarify that our research does not focus on the specific chemistries optimization of SSBs. Still, it focuses on providing the universal design principles to achieve high energy density for practical SSBs by systematic and multiscale approaches and fabrication methodologies, and those characteristics at the pouch cell level are for demonstrating our design guidelines. We were also convinced that the initial performance of our SSB pouch cell was enough to prove the effectiveness of our design logic, which was the first demonstration of Ah-scale and multi-stacked SSB pouch cell for scientific peer review. We do believe that the cycle characteristics of pouch cells are not relative to the proposed design strategies. This is because performance factors such as energy density and power density can be calculated, but the cycle characteristics are not predictable without recent big data-based machine learning tools.

Nevertheless, we admit that cyclability should be better for practical usage even if it is not in the scope of our research. The cycle performance of our cell slightly improves when operated at a higher C-rate than 0.05C (We added the data in **Supplementary Fig. 19**). The cell designed by our design logic can achieve several cycle numbers although it is a coin cell level. However, the capacity suddenly drops after about 150~160 hours because the dehydrofluorination of PVDF-HFP increases over time.

So, we tried to suppress the dehydrofluorination with several methods such as washing, annealing the cathode material, and lowering the operation temperature to counteract the effects of residual lithium compounds on the cathode surface and elevated temperature. The electrochemical performances from all tries were still suddenly dropped after a similar time passed. Therefore, we agree that the optimizing chemical compatibility is necessary to achieve a more extended cycle life and a longer operation time. However, given that the cell shows an improved cycle number (10 cycles) by controlling the charge/discharge protocol, our design logic is a proper guide for designing the electrode. Even though the cyclability is insufficient in this study, it is noteworthy that the energy density of pouch-type SSBs could be achieved well, similar to the calculated energy density based on the design logic considering the most ideal case. This study can provide insight into the design of the various parameters from electrodes to the cell fabrications and operating conditions to achieve the desired energy density of SSBs with various types of AM and SE with different chemistries.

Page 21, line 17

Original: However, this performance is still not comparable to that of LIBs, potentially because of a possible chemical side reaction, the well-known dehydrofluorination for PVDF-based polymers⁵⁰.

Revised: However, this performance is still not comparable to that of LIBs, potentially because of a possible chemical side reaction, the well-known dehydrofluorination for PVDF-based polymers⁵³

(Supplementary Fig. 3), and increased resistance because of high cathode loading weight and a large number of electrodes in the pouch-type stack. To suppress the dehydrofluorination, several methods such as washing, annealing the cathode material, and lowering the operation temperature to counteract the effects of residual lithium on the cathode surface and elevated temperature, were tried. The electrochemical performances from all tries are still suddenly dropped after approximately 150~160 hours. However, given that the cell shows an improved cycle number (10 cycles) by controlling the charge/discharge protocol, our design logic is a proper guideline for designing the electrode, as shown

in Supplementary Fig. 19.

Page 28, line 1 (Supplementary Information)

Supplementary Fig. 19: Cycle performance of Li/SPE/NCM622 coin cells under different C-rate protocols. 0.05C operation at 45°C with drying (a) and annealing (b) DI water of washed cathode, c, 0.1C charge / 0.2C discharge at RT with drying cathode, and d, electrochemical voltage vs. time profiles during cycling of three cases before the sudden capacity fading.

Notification for the additional refinement of the manuscript

In addition to addressing the specific points raised by the reviewers, we have also undertaken a thorough revision of the manuscript to refine and improve its overall quality. These edits include tightening the narrative for clarity, improving the organization of our arguments for better flow, and enhancing the presentation of our data for more impactful communication of our findings. We believe these modifications significantly augment the manuscript's contribution to the field.

Furthermore, we would like to inform the editor and the reviews of a minor change in the authorship and acknowledgment sections of our manuscript. These alterations reflect the evolving nature of collaborative research and our commitment to accurately representing contributions to the work.

1. Authorship Change: we would like to highlight a change in the authorship and contributions from the original submission to this revised version. These changes reflect the dynamic nature of our research and the evolving contributions of the team members.

- **Original Authorship:** Wonmi Lee, Taegyun Yu, Juho Lee, Sungbin Jang, Juhee Kim, Yujin Han, Sunghun Choi, Sinho Choi, Taehee Kim, Sang-Hoon Park, Wooyoung Jin, Gyujin Song, Daeil Kim, Bo-Yun Jang, Dong-Hwa Seo, Sung-Kyun Jung, Jinsoo Kim
- **Revised Authorship:** Wonmi Lee, Juho Lee, Taegyun Yu, Hyeong-Jong Kim, Min Kyung Kim, Sungbin Jang, Juhee Kim, Yujin Han, Sunghun Choi, Sinho Choi, Taehee Kim, Sang-Hoon Park, Wooyoung Jin, Gyujin Song, Dong-Hwa Seo, Sung-Kyun Jung, Jinsoo Kim
- **Addition of Authors:** Hyeong-Jong Kim and Min Kyung Kim have been added to the author list in recognition of their significant contributions to the experimental work and data analysis.
- **Author Order Change:** Juho Lee has been moved up in the author order to reflect his increased contribution in the experimental design and manuscript preparation.
- **Contributions Update:** The contributions section has been updated to accurately reflect the roles

and contributions of each author. Notably, Hyeong-Jong Kim's involvement in simulating the 3D digital-twin model and Min Kyung Kim's role in fabricating the SPE-based pouch cell have been added.

2. Acknowledgment Section Update: The acknowledgment section now additionally cites support from the National Research Foundation of Korea (NRF) grant funded by the Korea government (MSIT) (No. 2022R1C1C1006575 and 2023R1A2C2008242), acknowledging the broader funding support for our research.

We trust that these changes provide a clearer and more accurate representation of the contributions towards the manuscript. The revised manuscript, along with a detailed point-by-point response to the reviewers' comments, is attached for your consideration. We appreciate the opportunity to revise our manuscript and thank you for your continued consideration of our work for publication in *Nature Communications*.

Reference

1. Janek, J. & Zeier, W.G. Challenges in speeding up solid-state battery development. *Nat. Energy* **8**, 230-240 (2023).
2. Li, Q., Yang, Y., Yu, X. & Li, H. A $700 \text{ W} \cdot \text{h} \cdot \text{kg}^{-1}$ Rechargeable Pouch Type Lithium Battery. *Chin. Phys. Lett.* **40**, 048201 (2023).
3. Hong, C., Leng, Q., Zhu, J., Zheng, S., He, H., Li, Y., Liu, R., Wan, J., & Yang, Y. Revealing the correlation between structural evolution and Li^+ diffusion kinetics of nickel-rich cathode materials in Li-ion batteries. *J. Mater. Chem. A*, **8** 8540-8547 (2020)
4. Lee, M. J., Han, J., Lee, K., Lee, Y. J., Kim, B. G., Jung, K. N., Kim, B. J. & Lee, S. W. Elastomeric electrolytes for high-energy solid-state lithium batteries. *Nature* **601**, 217-222 (2022).
5. Chen, X., Sun, C., Wang, K., Dong, W., Han, J., Ning, D., Li, Y., Wu, W., Yang, C. & Lu, Z. An Ultra-Thin Crosslinked Carbonate Ester Electrolyte for 24 V Bipolar Lithium-Metal Batteries. *J. Electrochem. Soc.* **169**, 090509 (2022).
6. Ma, Y., Wan, J., Yang, Y., Ye, Y., Xiao, X., Boyle, D. T., Burke, W., Huang, Z., Chen, H., Cui, Y., Yu, Z., Oyakhire, S. T. & Cui, Y. Scalable, ultrathin, and high-temperature-resistant solid polymer electrolytes for energy-dense lithium metal batteries. *Adv. Energy Mater.* **12**, 2103720 (2022).
7. Zhao, X., Xiang, P., Wu, J., Liu, Z., Shen, L., Liu, G., Tian, Z., Chen, L. & Yao, X. Toluene tolerated $\text{Li}_9.88\text{GeP}_1.96\text{Sb}_0.04\text{S}_{11.88}\text{Cl}_{10.12}$ solid electrolyte toward ultrathin membranes for all-solid-state lithium batteries. *Nano Lett.* **23**, 227-234 (2022).
8. Seong, W. M., Cho, K. H., Park, J. W., Park, H., Eum, D., Lee, M. H., Kim, I. S., Lim, J. & Kang, K. Controlling residual lithium in high-nickel ($> 90\%$) lithium layered oxides for cathodes in lithium-ion batteries. *Angew. Chem. Int. Ed.* **59**, 18662-18669 (2020).
9. Taguet, A., Ameduri, B. & Boutevin, B. Crosslinking of vinylidene fluoride-containing fluoropolymers. In: Crosslinking in Materials Science. *Adv. Polym. Sci.* **184**, 127-211 (2005).
10. Tan, D. H., Meng, Y. S. & Jang, J. Scaling up high-energy-density sulfidic solid-state batteries: A

lab-to-pilot perspective. *Joule* **6**, 1755-1769 (2022).

11. Frith, J. T., Lacey, M. J. & Ulissi, U. A non-academic perspective on the future of lithium-based batteries. *Nat. Commun.* **14**, 1–17 (2023).
12. Shen, Y. Deng, G.-H. Ge, C. Tian, Y. Wu, G. Yang, X. Zheng, J. & Yuan, K. Solvation structure around the Li⁺ ion in succinonitrile–lithium salt plastic crystalline electrolytes. *Phys. Chem. Chem. Phys.* **18**, 14867-14873 (2016)

REVIEWER COMMENTS

Reviewer #1 (Remarks to the Author):

Dear editor and authors, I think the revised manuscript addresses most of my concerns. I agree that the point is not showing remarkable performance, but rather give a rational guidance for SSB design. I think the revised version provides this. I do have two comments: I've been unable to review the excel file, as in all devices I tried to access it, I got an error from excel saying the file is corrupted. Please, double check. However, the provided screenshots are a good indication of the spreadsheet functionality.

Secondly, I still disagree that SN is stable, there is no evidence provided by the authors that its decomposition forms a stable passivation layer on lithium metal. The authors only provided a single XPS spectra, and the provided citation (S2) is not relative to a study of the SEI upon time on lithium metal using this family of polymer electrolytes.

The authors should at least clearly state the limitation of their chemistry, clearly mention that it's not competitive, as it seems that both cathodic and anodic stability of their solid polymer electrolyte are very limited, and that's probably what's causing a very poor cycle life.

Reviewer #2 (Remarks to the Author):

This revised manuscript can be accepted without further revisions.

Response Letter

Manuscript NCOMMS-23-18937-B

Title: Ultimate design principles and fabrication process for high-energy-density solid-state pouch cells based on solid polymer electrolytes

Authors: Wonmi Lee, Juho Lee, Taegyun Yu, Hyeong-Jong Kim, Min Kyung Kim, Sungbin Jang, Juhee Kim, Yujin Han, Sunghun Choi, Sinho Choi, Taehee Kim, Sang-Hoon Park, Wooyoung Jin, Gyujin Song, Dong-Hwa Seo, Sung-Kyun Jung, and Jinsoo Kim

Reviewer #1

Dear editor and authors, I think the revised manuscript addresses most of my concerns. I agree that the point is not showing remarkable performance, but rather give a rational guidance for SSB design. I think the revised version provides this.

→ **Response:** We are thankful for the reviewer's positive assessment of our work. With the revision of our manuscript in the following, we hope that our conclusion has been further reinforced.

Continued from the comment of Reviwer #1:

1-1. I've been unable to review the excel file, as in all devices I tried to access it, I got an error from excel saying the file is corrupted. Please, double check. However, the provided screenshots are a good indication of the spreadsheet functionality.

→ **Response:** Thank you for your pointing out the error in the cell design spreadsheet. Regarding your comments on the Excel file, we apologize for the file access issue. Upon investigating this matter, we discovered that the root of the problem stemmed from the format conversion of the Excel file. Originally, the file was in "xlsm" format, embedded with VBA macros to automate certain functionalities. However, during the process through the Nature Publishing Group (NPG) submission system, it was converted to "xlsx" format, which unfortunately does not support macro functionality.

This conversion likely led to the error encountered when trying to access the file. To rectify this issue, in File Explorer, we recommend editing the file format from the converted "xlsx" to the original "xlsm" to ensure the embedded VBA macros function as intended. We have also verified that our SolidXCell operates correctly on both Windows and Mac operating systems. For users downloading the file from the internet, navigate to the file's properties via the General tab, and in the Security section, unblock the option that reads "This file came from another computer and might be blocked to help protect this computer." We hope that these clarifications and corrections address your concerns satisfactorily.

Page 29, line 6

Original:

Data availability

The data supporting the findings of this study are available in the paper and its Supplementary Information; further data are available from the corresponding author on request.

Revised:

Data availability

The data supporting the findings of this study are available in the paper and its Supplementary Information; further data are available from the corresponding author on request. **Supplementary Table**

1-3 for SSB cell design is available in the provided SolidXCell toolkit. After downloading the file, edit the Excel file format from 'xlsx' to 'xlsm' for functional VBA macro, and navigate to the Properties - General tab - Unblock the security option.

Continued from the comment of Reviewer #1:

1-2. Secondly, I still disagree that SN is stable, there is no evidence provided by the authors that its decomposition forms a stable passivation layer on lithium metal. The authors only provided a single XPS spectra, and the provided citation (S2) is not relative to a study of the SEI upon time on lithium metal using this family of polymer electrolytes.

The authors should at least clearly state the limitation of their chemistry, clearly mention that it's not competitive, as it seems that both cathodic and anodic stability of their solid polymer electrolyte are very limited, and that's probably what's causing a very poor cycle life.

→ **Response:** We appreciate your valuable comment. We agree with your opinion that our SPE is not sufficient due to some chemical side reactions, stemming from not only degradation of PVDF-HFP but also side reaction between SN and Li metal anode (i.e., polymerization of nitrile groups in SN, catalyzed by Li metal when they have direct contact)^{1,2}. Even though the LiF and Li₃N formation could be observed onto the Li metal anode based on XPS spectra, the interfacial stability between Li metal anode and SPE can be still problematic due to the serious side reaction between SN and Li metal. We added the sentence about the problem of SN with Li metal in the manuscript and mentioned that the SPE should be modified further to improve the cathodic and anodic stability for improving the cycle life, especially in terms of the chemistry perspective. In addition, we changed the citation of S2. We hope this explanation can relieve the reviewer's concern.

Page 21, line 17

Original: However, this performance is still not comparable to that of LIBs, potentially because of a possible chemical side reaction, the well-known dehydrofluorination for PVDF-based polymers⁵³

(Supplementary Fig. 3). and increased resistance because of high cathode loading weight and a large number of electrodes in the pouch-type stack. To suppress the dehydrofluorination, several methods

such as washing, annealing the cathode material, and lowering the operation temperature to counteract the effects of residual lithium on the cathode surface and elevated temperature, were tried. The electrochemical performances from all tries are still suddenly dropped after approximately 150~160 hours. However, given that the cell shows an improved cycle number (10 cycles) by controlling the charge/discharge protocol, our design logic is a proper guideline for designing the electrode, as shown in Supplementary Fig. 19. These problems require future work by the research community. Even so, we expect this to be a matter of unoptimized chemistries rather than cell design engineering.

Revised: However, this performance is still not comparable to that of LIBs, potentially because of a possible chemical side reaction, the well-known dehydrofluorination for PVDF-based polymers⁵³ (Supplementary Fig. 3) and the polymerization of nitrile groups in SN catalyzed by Li metal, incurring high interfacial resistance^{54,55}, and increased resistance because of high cathode loading weight and a large number of electrodes in the pouch-type stack. To suppress the dehydrofluorination, several methods such as washing, annealing the cathode material, and lowering the operation temperature to counteract the effects of residual lithium on the cathode surface and elevated temperature, were tried. The electrochemical performances from all tries are still suddenly dropped after approximately 150~160 hours. However, given that the cell shows an improved cycle number (10 cycles) by controlling the charge/discharge protocol, our design logic is a proper guideline for designing the electrode, as shown in Supplementary Fig. 19. Overall, this solid polymer electrolyte (SPE) requires further modification due to its limited cathodic and anodic stability, which results in a significantly poor cycle life. These problems require future work by the research community. Even so, we expect this to be a matter of unoptimized chemistries rather than cell design engineering.

Page 36, line 11

54. Zha, W., Li, J., Li, W., Sun, C. & Wen, Z. Anchoring succinonitrile by solvent-Li⁺ associations for high-performance solid-state lithium battery. *Chem. Eng. J.* **406** 126754 (2021).

55. Jeong, W., Kim, C., Kim, Y. J. & Lee, J. W. Stable lithium metal anode enabled by a polymeric interlayer with high salt concentration for solid-state batteries. *J. Energy Storage* **84** 110932 (2024).

Page 36, line 5 (Supplementary Information)

Original: S2. Park, S., Jeong, S. Y., Lee, T. K., Park, M. W., Lim, H. Y., Sung, J., Cho, J., Kwak, S. K., Hong, S. Y. & Choi, N. S. Replacing conventional battery electrolyte additives with dioxolone derivatives for high-energy-density lithium-ion batteries. *Nat. Commun.* **12**, 838 (2021).

Revised: S2. Zhang, Z., Wang, J., Zhang, S., Ying, H., Zhuang, Z., Ma, F., Huang, P., Yang, T., Han, G. & Han, W. Q. Stable all-solid-state lithium metal batteries with Li₃N-LiF-enriched interface induced by lithium nitrate addition. *Energy Storage Mater.* **43**, 229-237 (2021).

Reviewer #2

This revised manuscript can be accepted without further revisions.

→ **Response:** We deeply appreciate the reviewer's acceptance of our work for publication in this prestigious journal.

Reference

1. Zha, W., Li, J., Li, W., Sun, C. & Wen, Z. Anchoring succinonitrile by solvent-Li⁺ associations for high-performance solid-state lithium battery. *Chem. Eng. J.* **406** 126754 (2021).
2. Jeong, W., Kim, C., Kim, Y. J. & Lee, J. W. Stable lithium metal anode enabled by a polymeric interlayer with high salt concentration for solid-state batteries. *J. Energy Storage* **84** 110932 (2024).